# Neural ODE and SDE Models for Adaptation and Planning in Model-Based Reinforcement Learning

**Chao Han**[†]                                                   *chao.han@cruk.manchester.ac.uk*
*School of Computer Science, The University of Sheffield, UK*
*Present address: Cancer Research UK National Biomarker Centre, The University of Manchester, UK*

**Stefanos Ioannou**[†]                                                   *si386@cam.ac.uk*
*School of Computer Science, The University of Sheffield, UK*

**Luca Manneschi**                                                   *l.manneschi@sheffield.ac.uk*
*School of Computer Science, The University of Sheffield, UK*

**T.J. Hayward**                                                   *t.hayward@sheffield.ac.uk*
*School of Chemical, Materials and Biological Engineering, The University of Sheffield, UK*

**Michael Mangan**                                                   *m.mangan@sheffield.ac.uk*
*School of Computer Science, The University of Sheffield, UK*

**Aditya Gilra**                                                   *aditya.gilra@wu.ac.at*
*Machine Learning group, Centrum Wiskunde & Informatica, Amsterdam, Netherlands*
*School of Computer Science, The University of Sheffield, UK*
*Present address: Institute for Ecological Economics, Vienna University of Economics and Business, Austria*

**Eleni Vasilaki**                                                   *e.vasilaki@sheffield.ac.uk*
*School of Computer Science, The University of Sheffield, UK*

**Reviewed on OpenReview:** *https://openreview.net/forum?id=T6OrPlyPV4*

[†]Equal Contribution.

## Abstract

We investigate neural ordinary and stochastic differential equations (neural ODEs and SDEs) to model stochastic dynamics in fully and partially observed environments within a model-based reinforcement learning (RL) framework. Through a sequence of simulations, we show that neural SDEs more effectively capture transition dynamics' inherent stochasticity, enabling high-performing policies with improved sample efficiency in challenging scenarios. We leverage neural ODEs and SDEs for efficient policy adaptation to changes in environment dynamics via inverse models, requiring only limited interactions with the new environment. To address partial observability, we introduce a latent SDE model that combines an ODE and a GAN-trained stochastic component in latent space. Policies derived from this model offer a strong baseline, outperforming or matching general model-based and model-free approaches across stochastic continuous-control benchmarks. This work illustrates the applicability of action-conditional latent SDEs for RL planning in environments with stochastic transitions. Our code is available at: `https://github.com/ChaoHan-UoS/NeuralRL`.

## 1 Introduction

In recent years, the family of neural differential equations (neural DEs) (Chen et al., 2018; Rubanova et al., 2019; Li et al., 2020; Kidger et al., 2020; 2021) have emerged as a powerful framework for modelling dynamical

systems. The general idea of these models is to use neural networks to parameterise the derivatives of the system's dynamics, while the state evolution is computed by a numerical differential equation solver. Such a way of decoupling the modelling of dynamics from the discretisation scheme leads to models with increased fidelity for capturing complex, continuous-time transitions. As a result, neural DEs are especially well-suited for learning and representing transition dynamics in reinforcement learning (RL).

Recent work has demonstrated that integrating neural ordinary differential equations (neural ODEs) (Chen et al., 2018) and stochastic differential equations (neural SDEs) Li et al. (2020); Kidger et al. (2021) into RL frameworks can improve the modelling of continuous-time dynamics. In partially observable Markov decision processes (POMDPs) and model-free RL, recurrent neural ODEs have shown robustness to irregularly sampled observations, due to their capacity to model non-uniform time series (Zhao et al., 2023). Latent neural ODEs have also been employed as data-driven dynamics models in model-based RL, providing higher sample efficiency compared to model-free baselines (Du et al., 2020). Additionally, incorporating neural ODEs with control-theoretic constraints has supported the development of safe and stable RL in continuous-time domains (Zhao et al., 2025).

Neural SDEs extend the neural ODE framework by jointly modelling the deterministic and stochastic components of system dynamics, allowing for uncertainty-aware policy learning. Recent studies show that offline model-based RL with neural SDEs, particularly when incorporating physics priors, can outperform state-of-the-art algorithms on low-quality datasets (Koprulu et al., 2025). Physics-constrained neural SDEs, which explicitly represent model uncertainty, have also been shown to support generalisation beyond the training distribution (Djeumou et al., 2023). While both these neural SDE frameworks could, in principle, be applied to POMDPs, practical demonstrations to date have focused on environments with full observability.

Despite recent advances, the application of latent neural SDEs to partially observed, stochastic RL environments remains underexplored. Unified frameworks that combine controlled latent neural ODEs and neural SDEs in both fully and partially observed environments are lacking, and the practical benefits of modelling stochasticity in latent dynamics for planning have not been systematically assessed. Additionally, there has been limited exploration of neural ODEs and neural SDEs in settings that require adaptation to changes in the environment.

In this work, we first demonstrate sample-efficient policy adaptation to changes in environment configuration by employing an inverse dynamics approach (Christiano et al., 2016) based on neural ODE and SDE transition models. We then introduce a unified latent ODE/SDE framework for model-based RL, motivated by the need to handle partial observability in complex, stochastic environments. The framework uses a two-phase procedure: the mean latent dynamics are learned using a latent ODE, and stochasticity is handled via a GAN-trained latent SDE. In experiments on stochastic continuous control tasks, we show that model-based RL with latent neural SDEs improves sample efficiency compared to both ODE-based models and a model-free SAC baseline in the most challenging fully or partially observed environment included in our study.

To our knowledge, there has been no prior empirical validation of action-conditioned latent SDEs for planning in partially observed stochastic control tasks. Our results highlight the potential benefit of latent neural SDEs for flexible, noise-aware planning in stochastic domains.

## 2 Background

**Markov decision process (MDP) and partially observable MDP (POMDP).** An MDP (Cassandra et al., 1994) is defined by a tuple $\langle \mathcal{S}, \mathcal{A}, \mathcal{T}, R \rangle$, where $\mathcal{S}$ and $\mathcal{A}$ are sets of states and actions respectively, $\mathcal{T} : \mathcal{S} \times \mathcal{A} \to \Delta(\mathcal{S})$ is the probabilistic transition function (dynamics), with $\mathcal{T}(s_{t+1}|s_t, a_t)$ representing the probability of transitioning into $s_{t+1}$ from $s_t$ under $a_t$, $R : \mathcal{S} \times \mathcal{A} \times \mathcal{S} \to \mathbb{R}$ is the deterministic reward function. The initial state $s_0$ follows certain distribution $\rho_0$, the horizon is $T \in \mathbb{N}^+ \cup \{+\infty\}$ and the discount factor is $\gamma \in [0, 1]$, with 1 for finite horizon. When the state $s_t$ is not fully observable, MDP tuples are extended to POMDP (Cassandra et al., 1994) $\langle \mathcal{S}, \mathcal{A}, \mathcal{O}, \mathcal{T}, O, R \rangle$ by adding a set of observations $\mathcal{O}$ and an emission function $O : \mathcal{S} \to \Delta(\mathcal{O})$, which probabilistically maps a state $s_t$ to an observation $o_t$.

Specifically, we consider POMDPs with two sources of uncertainty in the transition dynamics of observed states $\mathcal{T}(o_{t+1} \mid o_t, a_t)$: (i) aleatoric stochasticity in the underlying MDP transition $\mathcal{T}(s_{t+1} \mid s_t, a_t)$ that

produces the observations, and (ii) epistemic uncertainty from partial observability, which can often be reduced by conditioning on a history of observations and actions. To focus on transition stochasticity and decouple it from aleatoric uncertainty in observation emission, we assume deterministic emission functions $O$, i.e., $o_t = o(s_t)$. Nonetheless, our approach readily generalizes to noisy observation emissions by replacing the deterministic observation model with a probabilistic neural network that outputs the statistics of the predicted observation distribution.

In such POMDPs, we introduce a latent representation $z_t$ that summarizes the observed history up to $t$ $\{a_{0:t-1}, o_{0:t}\}$ as a proxy of the state $s_t$ (Du et al., 2020; Ni et al., 2024). The transition function of the latent $z_t$ is $\mathcal{T}(z_{t+1}|z_t, a_t, o_t)$, with the observation $o_t$ deterministically emitted from $z_t$ via $o_t = o(z_t)$. We are interested in learning the encoder mapping the history to the latent variable, the latent transition function and the decoder mapping the latent back to the observation, which together enable the prediction of the next observation in the POMDP.

**Neural Ordinary Differential Equation (Neural ODE).** Neural differential equations are a family of differential equations whose rate functions are approximated by learnable neural networks (Kidger, 2022). A classic example is a neural ODE (Chen et al., 2018):

$$\frac{\mathrm{d}s_t}{\mathrm{d}t} = f_\theta(s_t, t), \quad \text{where} \quad s_{t_0} = s_0 \tag{1}$$

where $s_t$ denotes the state at time $t$ as the solution of the ODE initial-value problem (IVP). The initial state at $t_0$ is $s_0$. The rate function $f_\theta$ is mostly parameterized by a multi-layer perceptron (MLP) with learnable parameters $\theta$. The general regime of neural ODE allows any off-the-shelf numerical integrator to solve for the above ODE IVP. More concretely:

$$\hat{s}_{i+1} = \text{ODEsolve}(f_\theta(\hat{s}_i, t_0 + i\Delta t)), \quad i = 0, \ldots, T - 1 \tag{2}$$

where neural ODEsolve refers to the ODE numerical integrator used. $\hat{s}_{i+1}$ is a step forward of the neural ODEsolve from $\hat{s}_i$ with a pre-defined, fixed step size $\Delta t$.

A latent ODE (Rubanova et al., 2019) utilizes the neural ODE to generate latent trajectories in a variational autoencoder (VAE) manner (Kingma, 2013), which can be formulated as:

$$z_0 \sim q_\phi(z_0|o_{0:T}), \quad z_{i+1} = \text{ODEsolve}(f_\theta(z_i, t_0 + i\Delta t)), \quad \hat{o}_i = o_\theta(z_i), \quad i = 0, \ldots, T - 1 \tag{3}$$

Here, an (RNN) encoder parameterized by $\phi$ maps observations $o_{0:T}$ to a distribution of the initial latent state $z_0$. A decoder parameterized by $\theta$ reconstructs observations $\hat{o}_{0:T}$ via a latent-variable ODE determined by sampled $z_0$ and an emission function $o_\theta(\cdot)$.

**Neural Stochastic Differential Equation (Neural SDE).** A neural SDE consists of a parameterized deterministic drift term and a parameterized stochastic diffusion term:

$$\mathrm{d}s_t = f_\theta(s_t, t)\mathrm{d}t + g_\theta(s_t, t)\mathrm{d}w_t, \quad \text{where} \quad s_{t_0} \sim \mu \tag{4}$$

where $\{s_t\}_{t \geq t_0}$ is a continuous-time stochastic process, whose initial state $s_{t_0}$ is drawn from some probability distribution $\mu$. $\{w_t\}_{t \geq 0}$ is the Brownian motion. The paper follows the idea of training a neural SDE as a (Wasserstein) generative adversarial net (GAN) (Goodfellow et al., 2014; Arjovsky et al., 2017), where the real and generated data samples are (interpolated) infinite-dimensional paths (Kidger et al., 2021).

## 3 Methodology

In this section, we extend the vanilla autonomous neural ODE/SDE-based models to their controlled variants by incorporating actions to model the MDP state transition dynamics $\mathcal{T}(s_{t+1}|s_t, a_t)$ and POMDP latent transition $\mathcal{T}(z_{t+1}|z_t, a_t, o_t)$. We also describe how to train a policy for planning based on the learned transition model, as well as how to adapt the model and policy to a similar environment without retraining from scratch.

### 3.1 Dynamics model learning

We use the terms neural ODE/SDE and latent ODE to overload the names of original autonomous models and here denote their controlled variants. In a similar vein to latent neural ODEs, we also propose the latent SDE by employing the neural SDE in the encoded latent space, which, to the best of our knowledge, has not been proposed in the literature.

**Neural ODE.** Let the MDP transition dynamics of a deterministic environment be defined as an autonomous neural ODE ($t$ is not explicitly given as an argument to $f_\theta$):

$$\hat{s}_{i+1} = \text{ODEsolve}(f_\theta(\hat{s}_i, a_i)), \quad i = 0, \dots, T-1 \tag{5}$$

We optimize the parameters $\theta$ by minimizing the following mean squared error (MSE) between the predicted $\hat{s}_{0:T}$ and observed state trajectories $s_{0:T}$:

$$\min_\theta \mathbb{E}_{(s_{0:T}, a_{0:T-1}) \sim \mathcal{D}_m} \left[ \sum_{i=0}^{T} \|\hat{s}_i - s_i\|_2^2 \right] \tag{6}$$

where real state-action trajectories are sampled from a replay buffer $\mathcal{D}_m$.

**Latent ODE.** Similar to Du et al. (2020), we modify the vanilla latent ODE in Eq. 3 to model the POMDP latent transition by taking actions $a_i$ and (predicted) observations $\tilde{o}_i$ into account for the underlying evolution of RNN hidden states $h_i$ in the encoder and ODE latent states $z_i$ in the decoder. Specifically, the adjusted latent ODE consists of the following RNN encoder parameterized by $\phi$:

$$h_{i+1} = \text{RNNCell}_\phi(h_i, a_i, \tilde{o}_i), \quad \hat{o}_i = o_\phi(h_i), \quad i = 0, \dots, T-1,$$
$$[\mu_{z_0}, \sigma_{z_0}] = \text{MLP}_\phi(h_T), \quad q_\phi(z_0 | o_{0:T}, a_{0:T-1}) = \mathcal{N}(z_0; \mu_{z_0}, \text{diag}(\sigma_{z_0}^2)) \tag{7}$$

and neural ODE decoder parameterized by $\theta$:

$$z_0 \sim q_\phi(z_0 | o_{0:T}, a_{0:T-1}), \quad \tilde{z}_i = \text{MLP}_\theta(z_i, a_i, \tilde{o}_i), \quad z_{i+1} = \text{ODEsolve}(f_\theta(\tilde{z}_i)),$$
$$\hat{o}_i = o_\theta(z_i), \quad p_\theta(o_{i+1} | z_i, a_i, \tilde{o}_i) = \mathcal{N}(o_{i+1}; o_\theta(z_{i+1}), I), \quad i = 0, \dots, T-1 \tag{8}$$

where $o_\theta(\cdot)$ denotes the parameterized emission function, $\tilde{z}_i$ has the same dimension as $z_i$ and represents the transformed latent state at which the derivative of latent ODE $f_\theta$ is evaluated. $\tilde{o}_i$ could be either the real observation $o_i$ or predicted observation $\hat{o}_i$ during training, while always setting $\tilde{o}_i = \hat{o}_i$ during inference. In this paper, models are typically trained using teacher forcing, i.e., $\tilde{o}_i = o_i$, as this has been shown to be more sample-efficient and effective for training.

The above latent ODE is trained end-to-end by maximizing the evidence lower bound (ELBO) over observation trajectories $o_{0:T}$:

$$\max_{\phi, \theta} \mathbb{E}_{(o_{0:T}, a_{0:T-1}) \sim \mathcal{D}_m} \left[ \mathbb{E}_{z_0 \sim q_\phi(z_0 | o_{0:T}, a_{0:T-1})} \left[ \sum_{i=0}^{T-1} \log p_\theta(o_{i+1} | z_i, a_i, \tilde{o}_i) \right] \right.$$
$$\left. - D_{\text{KL}} \left( q_\phi(z_0 | o_{0:T}, a_{0:T-1}) \, \| \, p(z_0) \right) \right] \tag{9}$$

Here, the joint generative distribution is decomposed as $p_\theta(o_{0:T} | z_{0:T}, a_{0:T-1}) = \prod_{i=0}^{T} p_\theta(o_{i+1} | z_i, a_i, \tilde{o}_i)$ and $p(z_0) = \mathcal{N}(z_0; 0, I)$ is the standard Gaussian prior.

**Neural SDE.** We employ a GAN-based SDE to generate synthetic trajectories that closely match those sampled from a stochastic MDP transition. More precisely, the adapted neural SDE trained as Wasserstein GAN (WGAN) consists of a neural SDE generator parameterized by $\theta$:

$$\hat{s}_{i+1} = \text{SDEsolve}(f_\theta(\tilde{s}_i, a_i), g_\theta(\tilde{s}_i, a_i), \Delta w_i), \quad \bar{s}_i = \text{MLP}_\theta(\hat{s}_i) \quad i = 0, \dots, T-1 \tag{10}$$

and an MLP critic (discriminator) parameterized by $\psi$:

$$y = \text{MLP}_\psi(\bar{s}_{0:T}, a_{0:T-1}) \tag{11}$$

Here, the projection from $\hat{s}_i$ to $\bar{s}_i$ allows for more flexible generated states. Different from Kidger et al. (2021) that uses a neural controlled differential equation (CDE) as a discriminator, we simply use an MLP that takes flattened state-action trajectories as input. This change makes optimization much easier, but also limits the discriminator's capacity for processing varying-length trajectories.

Let $G_\theta : (w_{0:T}, a_{0:T-1}) \to s_{0:T}$ denote the overall map of the generator from paths of noises and actions to that of synthetic observations, and $D_\psi : (s_{0:T}, a_{0:T-1}) \to y$ denote the overall map of the critic from paths of real/synthetic observations and actions to a scalar score. The training dynamics of the WGAN are formulated as follows:

$$\min_\theta \max_{\substack{\psi \\ \|D_\psi\|_L \leq 1}} \mathbb{E}_{(s_{0:T}, a_{0:T-1}) \sim \mathcal{D}_m}[D_\psi(s_{0:T}, a_{0:T-1})] - \mathbb{E}_{\substack{w_{0:T} \sim p_w \\ a_{0:T-1} \sim \mathcal{D}}}[D_\psi(G_\theta(w_{0:T}, a_{0:T-1}), a_{0:T-1})] \tag{12}$$

where $\|D_\psi\|_L \leq 1$ represent the Lipschitz continuity constraint enforced on the critic function $D_\psi$. We use the gradient penalty (Gulrajani et al., 2017) to achieve the Lipschitz constraint.

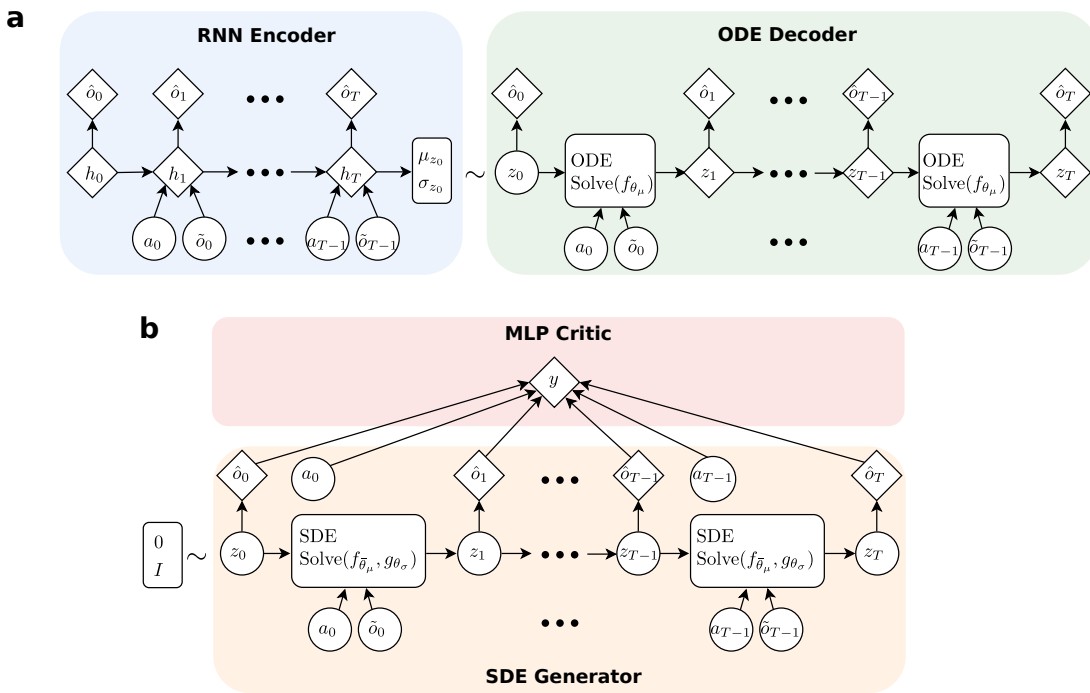

Figure 1: Computational graph of the latent SDE. **(a)** The RNN encoder and ODE decoder from the latent ODE. The encoder is only utilized during the model training. In the inference (generation) stage, the initial latent $z_0$ is sampled from the standard Gaussian prior distribution instead. Note that the $\tilde{o}_i$ could denote either the real observation $o_i$ or predicted one $\hat{o}_i$ during training, depending on whether teacher forcing strategy is used, yet it is always set to be $\tilde{o}_i = \hat{o}_i$ during inference. **(b)** The SDE generator and MLP critic of the latent SDE. In the SDE solver, we use the learnt drift function $f_{\bar{\theta}_\mu}$ from the ODE decoder in **(a)** and train the diffusion function $g_{\theta_\sigma}$ to capture the full stochasticity in the latent space. In both panels, the deterministic variables are represented as diamonds while the stochastic variables are depicted as circles.

**Latent SDE.** A weakness of latent ODE is that it can only model deterministic MDP transitions (or the mean dynamics of a stochastic MDP transitions) of states underlying a POMDP. To learn the full stochastic MDP state transition, we employed a phased training framework. The latent ODE (Eqs. 7 and 8) is first

employed to capture the mean transition dynamics in the latent space. Specifically, the encoder-decoder pair of the latent ODE is described as follows:

$$z_0 \sim q_\phi(z_0|o_{0:T}, a_{0:T-1}), \quad z_{i+1} = \text{ODEsolve}(f_{\theta_\mu}(z_i, a_i, \tilde{o}_i)), \quad \hat{o}_i = o_{\theta_\mu}(z_i), \quad i = 0, \ldots, T-1 \quad (13)$$

where the recurrent encoder and the ODE decoder are respectively parameterized by $\phi$ and $\theta_\mu$, which are trained in an end-to-end VAE-style setup (Eq. 9).

We then freeze the learned ODE decoder and deploy it as the drift function of an SDE in the latent space. The diffusion function of the SDE and the emission function are further employed to model the variance of the latent transition dynamics and the latent-to-observation projection, respectively. Putting together, we have the following equations:

$$z_0 \sim p(z_0), \quad z_{i+1} = \text{SDEsolve}(f_{\bar{\theta}_\mu}(z_i, a_i, \tilde{o}_i), g_{\theta_\sigma}(z_i, a_i, \tilde{o}_i), \Delta w_i) \quad \hat{o}_i = o_{\theta_\sigma}(z_i), \quad i = 0, \ldots, T-1,$$
$$y = \text{MLP}_\psi(\hat{o}_{0:T}, a_{0:T-1}) \quad (14)$$

where $z_0$ is sampled from the Gaussian prior $p(z_0)$ used in the VAE training. $f_{\bar{\theta}_\mu}$ and $g_{\theta_\sigma}$ are the drift and diffusion function of the SDE, where $\bar{\theta}_\mu$ denotes the detached optimal $\theta_\mu$ from the first phase. The diffusion $g_{\theta_\sigma}$ and emission function $o_{\theta_\sigma}$ of the SDE generator are trained in pairs with the MLP critic parameterized by $\psi$ in GAN-style setup (Eq. 12).

We term this model framework that combines latent ODE and neural SDE as latent SDE, and depict its computation graph in Fig. 1.

## 3.2 Planning and policy learning with learned dynamics

When addressing sample-intensive tasks (such as Mujoco), a model-free RL method is often trained off-policy using near-optimal transitions collected by a planning strategy driven by learned, task-specific dynamics, thereby reducing sample complexity. In this paper, following Du et al. (2020), we adopt a framework that combines model predictive control (MPC) (Nagabandi et al., 2017; Chua et al., 2018), which searches for the most promising exploratory action, with soft actor-critic (SAC) (Haarnoja et al., 2018), which learns the optimal policy (actor) and value function (critic).

At each time step $t$, MPC simulates $K$ trajectories over a planning horizon $H$, where actions and next states are sampled from the actor and the transition model, respectively. To prevent the model rollouts from becoming shortsighted, the critic estimates the cumulative reward beyond the planning horizon. This estimate, together with the cumulative reward from the MPC rollouts, forms the return for the state at time $t$. Subsequently, the first action of the trajectory that achieves the highest return is selected to interact with the environment.

This approach interleaves training of the transition model (when needed), data collection, and optimization of both the policy and the value function, thereby enabling policy learning from a limited number of data samples. The complete algorithm for POMDPs, adapted from Du et al. (2020), is provided in Algorithm 1, with the MPC planning component highlighted in blue.

## 3.3 Model and Policy Adaptation

In this subsection, we introduce an efficient way to adapt the learned transition model and policy of a source environment to a target environment, at the cost of a minimal amount of data from the target environment. The source environment typically refers to a simulated or controlled setting while the target environment denotes the real-world or modified simulated scenario, usually presenting novel dynamics or disturbances. Inspired by prior work by Christiano et al. (2016), we employ an adaptation architecture based on an inverse dynamics model, which allows us to leverage the high-level characteristics of the source policy while adapting to the specifics of the target domain.

In what follows, we use the superscripts "src" and "tge" to denote variables and models associated with the source and target environments, respectively. We assume that both environments are characterized by MDP

---

**Algorithm 1:** MPC with SAC for POMDPs.

---

**Input** : The empty replay buffer for model learning $\mathcal{D}_m$ and for policy learning $\mathcal{D}_p$, the initial latent transition model $\mathcal{T}_\theta(z_{t+1}|z_t, a_t, o_t)$, the reward function $R(o_t, a_t, o_{t+1})$, the initial actor $\pi_\omega(a_t|o_t)$, the two critics $Q^\pi_{\varphi_i}(o_t, a_t)$ and their target networks $Q^\pi_{\varphi'_i}(o_t, a_t)$, $i = 1, 2$.

**Params:** The planning horizon $H$, the search population $K$, the number of environment steps $M$ and the number of epochs $E$.

**Output:** Learned $P_\theta$, $\pi_\omega$ and $\min_{i=1,2} Q^\pi_{\varphi_i}$.

**1** Collect trajectories using a uniformly distributed policy and save them into $\mathcal{D}_m$;
**2** **for** $i = 1$ **to** $E$ **do**
**3**     Update $\mathcal{T}_\theta(z_{t+1}|z_t, a_t, o_t)$ using data from $\mathcal{D}_m$, as detailed in Section 3.1;
**4**     Observe the initial state $o_0$ and initialize the latent state $z_0$ ;
**5**     **for** $j = 1$ **to** $M$ **do**
**6**        **for** $k = 1$ **to** $K$ **do**
**7**           $\hat{o}^{(k)}_0, z^{(k)}_0 \leftarrow o_{j-1}, z_{j-1}$;
**8**           **for** $h = 1$ **to** $H$ **do**
**9**              Select the action $a^{(k)}_{h-1} \sim \pi_\omega(\cdot|\hat{o}^{(k)}_{h-1})$;
**10**              Transit to next latent $z^{(k)}_h \leftarrow \mathcal{T}_\theta(\cdot|z^{(k)}_{h-1}, a^{(k)}_{h-1}, \hat{o}^{(k)}_{h-1})$ and emit the observation $\hat{o}^{(k)}_h \leftarrow o_\theta(z^{(k)}_h)$;
**11**              Calculate the reward $\hat{r}^{(k)}_{h-1} \leftarrow R(\hat{o}^{(k)}_{h-1}, a^{(k)}_{h-1}, \hat{o}^{(k)}_h)$;
**12**           **end**
**13**           Select the action $a^{(k)}_H \sim \pi_\omega(\cdot|\hat{o}^{(k)}_H)$;
**14**        **end**
**15**        Select the index of the sequence with the maximum return among the $K$ sequences:
          $k^* \leftarrow \arg\max_k \sum^H_{h=1} \gamma^{h-1}\hat{r}^{(k)}_{h-1} + \gamma^H \min_{i=1,2} Q^\pi_{\varphi_i}(\hat{o}^{(k)}_H, a^{(k)}_H)$;
**16**        Select the first action of the best sequence $a_{j-1} \leftarrow a^{(k^*)}_0$;
**17**        Execute $a_{j-1}$ and observe the next state $o_j$;
**18**        Transit to the next latent $z_j \leftarrow \mathcal{T}_\theta(\cdot|z_{j-1}, a_{j-1}, o_{j-1})$;
**19**        Calculate the reward $r_{j-1} \leftarrow R(o_{j-1}, a_{j-1}, o_j)$;
**20**        **if** $o_j$ *is not the terminal state* **then**
**21**           Store $(o_{j-1}, a_{j-1}, r_{j-1}, o_j)$ in $\mathcal{D}_p$;
**22**        **else**
**23**           Store $(o_{j-1}, a_{j-1}, r_{j-1}, \text{NULL})$ in $\mathcal{D}_p$, reset the environment;
**24**           Observe the initial state $o_0$ and initialize the latent state $z_0$ ;
**25**           **continue**;
**26**        **end**
**27**        Update $Q^\pi_{\varphi_1}$, $Q^\pi_{\varphi_2}$ and $\pi_\omega$ using data from $\mathcal{D}_p$;
**28**        Update target networks $Q^\pi_{\varphi'_1}$, $Q^\pi_{\varphi'_2}$;
**29**     **end**
**30**     Store the trajectories collected using the current actor in $\mathcal{D}_m$;
**31** **end**

---

transitions and share the same actuated degrees of freedom. As illustrated in Fig. 2A, given a current target state $s^{\text{tge}}_t$, we first compute the corresponding source action $a^{\text{src}}_t = \pi^{\text{src}}(s^{\text{tge}}_t)$ using the predefined source policy $\pi^{\text{src}}$. Next, we estimate the subsequent source state $\hat{s}^{\text{src}}_{t+1} = \mathcal{T}^{\text{src}}_\theta(s^{\text{tge}}_t, a^{\text{src}}_t)$ via the parameterized source transition model $\mathcal{T}^{\text{src}}_\theta$, conditioned on the current state $s^{\text{tge}}_t$ and action $a^{\text{src}}_t$. Finally, the current target action $a^{\text{tge}}_t = I_\eta(s^{\text{tge}}_t, \hat{s}^{\text{src}}_{t+1})$ is computed using the inverse dynamics model $I_\eta$, which maps the current target state $s^{\text{tge}}_t$ and the desired next state $\hat{s}^{\text{src}}_{t+1}$to a target action that drives the next target state towards the desired state $\hat{s}^{\text{src}}_{t+1}$ as close as possible.

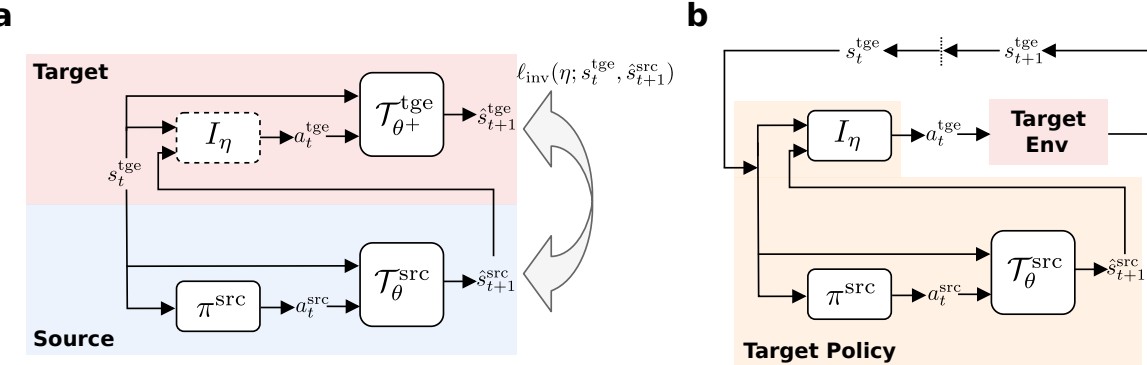

Figure 2: Overview of the architecture for training and deploying a policy adapted from the source domain in the target domain. **(a)** Given an off-the-shelf source policy $\pi^{\text{src}}$ and transition model $\mathcal{T}_\theta^{\text{src}}$, we use an inverse dynamics model $I_\eta$ to generate the target action $a_t^{\text{tge}}$ leading to the next target state $\hat{s}_{t+1}^{\text{tge}}$. $I_\eta$ is trained via minimizing the mismatch between the predicted next state $\hat{s}_{t+1}^{\text{tge}}$ and the desired state $\hat{s}_{t+1}^{\text{src}}$. **(b)** The trained inverse model, combined with the source policy and transition, is deployed in the real target environment as an adapted target policy.

**Data collection/training.** In order to minimize the mismatch between the next target state led by the target action and the anticipated next state, we need a differentiable transition model for the real transition dynamics in the target domain. Specifically, we adapt the deterministic component of the source transition model parameterized by $\theta$ to that of the target transition model parameterized by the augmented parameter set $\theta^+ = \theta \cup \theta_{\text{aug}}$, which includes additional parameters $\theta_{\text{aug}}$ that capture the variation in the deterministic component of transition dynamics between the source and target environments. We train only the additional parameters $\theta_{\text{aug}}$ of the augmented neural network that models the deterministic component of the target transition, using a small dataset collected under a random policy from the target environment and stored in the buffer $\mathcal{D}_m$. Similar to the loss for a deterministic Neural ODE defined in Eq. 6, we minimize the following MSE loss between the predicted and real target state trajectories:

$$\mathcal{L}_{\text{m}}(\theta_{\text{aug}}; s_{0:T}^{\text{tge}}, a_{0:T-1}^{\text{tge}}) = \mathbb{E}_{(s_{0:T}^{\text{tge}}, a_{0:T-1}^{\text{tge}}) \sim \mathcal{D}_m} [\ell_{\text{m}}(\theta_{\text{aug}}; s_{0:T}^{\text{tge}}, a_{0:T-1}^{\text{tge}})] \tag{15}$$

where

$$\ell_{\text{m}}(\theta_{\text{aug}}; s_{0:T}^{\text{tge}}, a_{0:T-1}^{\text{tge}}) = \sum_{t=1}^{T} \|\hat{s}_t^{\text{tge}} - s_t^{\text{tge}}\|_2^2 = \sum_{t=1}^{T} \|\mathcal{T}_{\theta^+}^{\text{tge}}(s_{t-1}^{\text{tge}}, a_{t-1}^{\text{tge}}) - s_t^{\text{tge}}\|_2^2 \tag{16}$$

Empirically we only require a small amount of data from the target domain to train the target transition model augmented from the source transition model, compared with training a target transition model from scratch. For example, in the cartpole adaptation task described in section 4.2, training a neural ODE from scratch requires 500k transitions, whereas fine-tuning only the final layer of a source-trained model achieves similar validation loss with just 2k target transitions. We freeze the target transition model once it is sufficiently accurate, and then use it for training the inverse dynamics model, which is interleaved with data collection. Specifically, we repeat the following data collection/training loop: during the data collection phase, given a current target state, we use the preliminary source policy and transition model to obtain the desired next state, and then use the learned-so-far inverse dynamics model to produce the target action, which interacts with the real target environment leading to the next target state. We repeat the target environment steps and save the sequence along the steps $\{\cdots, s_t^{\text{tge}}, \hat{s}_{t+1}^{\text{src}}, s_{t+1}^{\text{tge}}, \hat{s}_{t+2}^{\text{src}}, \cdots\}$ into the buffer $\mathcal{D}_{inv}$. Target states along the trajectory collected in such an on-the-fly fashion are near the optimal target trajectory, which will lead to much faster convergence of the inverse dynamics model compared with training data collected by a random policy. During the training phase, we optimize the following mean squared error (MSE) loss between the predicted next target state and the desired next state, using a mini-batch of

real-desired transition pairs sampled from $\mathcal{D}_{inv}$:

$$\mathcal{L}_{\text{inv}}(\eta; s_t^{\text{tge}}, \hat{s}_{t+1}^{\text{src}}) = \mathbb{E}_{(s_t^{\text{tge}}, \hat{s}_{t+1}^{\text{src}}) \sim \mathcal{D}_{inv}}[\ell_{\text{inv}}(\eta; s_t^{\text{tge}}, \hat{s}_{t+1}^{\text{src}})] \tag{17}$$

where

$$\ell_{\text{inv}}(\eta; s_t^{\text{tge}}, \hat{s}_{t+1}^{\text{src}}) = \|\hat{s}_{t+1}^{\text{tge}} - \hat{s}_{t+1}^{\text{src}}\|_2^2 = \|\mathcal{T}_{\theta^+}^{\text{tge}}(s_t^{\text{tge}}, I_\eta(s_t^{\text{tge}}, \hat{s}_{t+1}^{\text{src}})) - \hat{s}_{t+1}^{\text{src}}\|_2^2 \tag{18}$$

It is worth noting that the above MSE loss not only applies to the next states generated by deterministic transition dynamics but also to the means of the next states when the transition dynamics are stochastic. The averaged dynamics of the stochastic transition described by an SDE (i.e., the drift term of the SDE) can be captured by the Neural ODE model. Therefore we are only interested to adapt the model and policy when the mean-field (drift) dynamics vary between the source and target domain as the diffusion dynamics capture the aleatoric uncertainty, which is irreducible (Chua et al., 2018; Han et al., 2024).

**Deployment.** As illustrated in Fig. 2B, we can use the learned inverse dynamics model as a target policy, which is computed via adaptation from the source policy as follows:

$$\pi^{\text{tge}}(s_t^{\text{tge}}) = I_\eta(s_t^{\text{tge}}, \mathcal{T}_\theta^{\text{src}}(s_t^{\text{tge}}, \pi^{\text{src}}(s_t^{\text{tge}}))) \tag{19}$$

## 4 Experiments

We empirically evaluate our ODE- and SDE-based models in stochastic environments of increasing complexity. In section 4.1, we show that the neural SDE captures stochastic transitions more accurately than the neural ODE, which only learns the deterministic drift corresponding to the averaged dynamics. In section 4.2, we demonstrate that neural ODE/SDE models enable policy adaptation to environmental changes via the inverse model with fewer data. Finally, in section 4.3, we show that SDE-based policies generally achieve higher asymptotic rewards and faster convergence than other model-based and model-free baselines, under stochastic environments with full and partial observability.

**Environments.** For evaluation of our methods, we modify the standard deterministic OpenAI Gym environments (Towers et al., 2024): cartpole from the classic control task, swimmer, hopper and walker2d from the Mujoco locomotion task into stochastic versions by adding noises to their MDP transition dynamics. In each modified environment, both the action and observation spaces are continuous. When fully observable, the observation space includes the positions (or angles) and velocities (or angular velocities) of every degree of freedom.

- **Stochastic cartpole.** The goal is to balance a pole on a moving cart by applying forces on the cart at each step. We convert the original discrete actions to continuous ones and apply independent and identically distributed (i.i.d.) standard Gaussian force to the cart at each step, which can be formulated as an SDE of cart velocity.

- **Stochastic swimmer.** A 3-link swimmer is propelled forward in a fluid, with a 10-dimensional observation space and 2 actuators. We introduce i.i.d. Gaussian noise sampled from $\mathcal{N}(0, 500^2)$ to the stiffness parameter of the first actuated joint of the Swimmer.

- **Stochastic hopper.** A single-legged robot is made to hop forward as far as possible, with a 12-dimensional observation space and 3 actuators. We apply stochastic winds to the hopper at each step. The magnitude of the wind parameter is i.i.d. sampled from $\mathcal{N}(0, 5^2)$.

- **Stochastic walker2d.** A bipedal robot, characterized by an 18-dimensional observation space and 6 actuators, is tasked with walking forward while subjected to stochastic winds. The wind's magnitude is sampled i.i.d. at each step from $\mathcal{N}(0, 5^2)$.

We further hide the position and velocity features from the observation space of the Mujoco to evaluate learning in POMDPs. It is worth noting that the POMDPs considered here satisfy our assumptions on the aleatoric and epistemic nature of the uncertainties in the POMDP transition dynamics, as described in 2. Specifically, in the partially observable stochastic swimmer, we mask the positions and angle of the front tip (**stochastic swimmer (no position)**), and its positional and angular velocities (**stochastic swimmer (no velocity)**) from the observation space (corresponding to the first three and the 5th to 8th dimensions of the observation respectively). In the partially observable stochastic hopper, we mask the positions of the torso (**stochastic hopper (no position)**), and its angular velocity (**stochastic hopper (no velocity)**), which correspond to the first two and the 8th dimensions of the observation, respectively. Similarly, for the partially observable stochastic walker2d, we hide the torso's positional information in the first two dimensions (**stochastic walker2d (no position)**) and its angular velocity in the 11th dimension of the observation space (**stochastic walker2d (no velocity)**).

**Baselines.** We compare ODE-based and SDE-based models in stochastic environments with respect to both dynamics learning and model-based policy optimization. In the simpler CartPole task, we evaluate neural ODE (**N-ODE**) and neural SDE (**N-SDE**) as proxies for the true stochastic transition dynamics, testing their effectiveness for policy learning and adaptation. In the more challenging MuJoCo tasks, we additionally consider latent ODE (**L-ODE**) and latent SDE (**L-SDE**) models for planning and policy optimization. Across all environments, we use Soft Actor-Critic (**SAC**) (Haarnoja et al., 2018) as a model-free baseline, training Gaussian policies but evaluating their deterministic mean. As the model-based counterpart, we adopt Model-Based Policy Optimization (**MBPO**) (Janner et al., 2019), which improves sample efficiency by training on short-horizon imaginary rollouts generated from an ensemble of dynamics models, where each model is a probabilistic neural network parameterizing a Gaussian distribution (Chua et al., 2018; Janner et al., 2019).

It is worth noting that, although for computational efficiency we set the integrator step size equal to the simulator time step, the ODE/SDE-based models can in principle achieve finer temporal resolution by using smaller integrator steps. This added error control in time-series modeling is not available to general model-based RL baselines such as MBPO (Janner et al., 2019), which operate only on discrete-time MDPs with a fixed step size. A detailed experimental setup is provided in the Appendix C.

## 4.1 Neural ODE/SDE modelling transition dynamics

Here we demonstrate the capacity of the neural ODE and SDE in mimicking the stochastic MDP transitions of the stochastic cartpole environment. We use the learned transition models as a proxy for the real transition dynamics. A model-free agent can therefore be trained in the approximated transition dynamics without interacting with the real environment. To achieve the oracle performance given by the agent trained in the real environment, the transition model should be accurate enough to recover the distribution of the possible next states.

**Neural ODEs model the mean of stochastic transitions while Neural SDEs capture the full stochastic dynamics.** Fig. 3a depicts the evolution of marginal distributions of cart velocity across time predicted by the neural ODE and SDE, in comparison with the real marginal distribution. The ODE-based distributions are peakier at the mean than the real distribution and fail to recover values away from the mean. On the other hand, the SDE-based distributions mostly cover the real distributions. In addition, Fig. 3b shows the sample paths from the ODE and SDE-based distributions against the paths from the real distributions. Once again the ODE-based paths capture only the averaged tendency of the real paths, while SDE-based paths show better agreement with the real ones. For other features in the observation space of the stochastic cartpole, since their transition dynamics are deterministic, the neural SDE learns almost the same transition dynamics as the neural ODE (see Fig. 6 and 7 for distribution and path matching respectively in the Appendix A).

We evaluate the performance of agents trained on modeled transition dynamics in the real environment (Fig. 3c). Performance of SDE-based policy is much closer to the near-optimal performance of the model-free oracle policy than the ODE-based policy, in terms of the resemblance of their distribution of returns to the

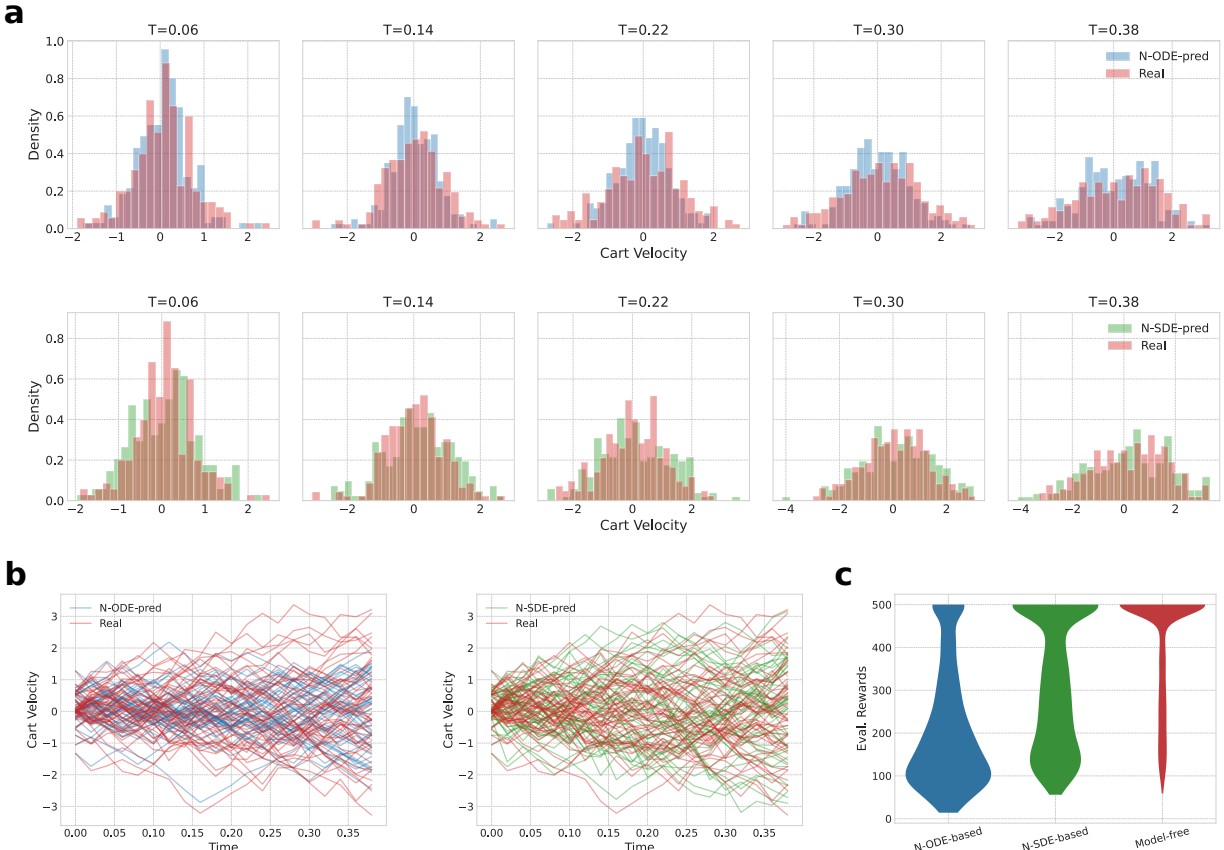

Figure 3: Comparison between neural ODE and SDE on stochastic cartpole. **(a)** Predicted histograms of cart velocity at 16%, 37%, 58%, 79%, 100% of the total timesteps by the neural ODE and SDE against the real histograms. **(b)** 50 sample paths from the distributions predicted by neural ODE and SDE against 50 paths from the real distributions. The neural SDE learns to match the real distributions and sample paths better than the neural ODE. **(c)** Policies trained in the approximate transition dynamics of neural ODE (N-ODE-based) and SDE (N-SDE-based), as well as in real transition (model-free), are evaluated in the real environment. N-SDE-based policy achieves similar performance to the model-free policy (used as the oracle here).

oracle distribution. The poor performance of ODE-based policy when deployed in the actual environment is due to the low fidelity of the model used for training the policy.

## 4.2 Policy adaptation via inverse dynamics model

In this section, we empirically test the policy adaptation framework described in section 3.3 in both deterministic and stochastic cartpole environments with increasing pole length. We compare different transition models: the neural ODE, the neural SDE, and the ensemble of Gaussian neural networks. The results show that adapted source policies based on these transition models outperform non-adapted source policies in the target environment, while also achieving greater sample efficiency than training new target policies from scratch under limited interaction with the target environment.

**Deterministic transition.** We first consider the setting where the cartpole dynamics are deterministic (Fig. 4a). We compare two adapted policies against the non-adapted source policy (red curve) as the pole length increases. The adapted policies are based on neural ODEs (blue curve) and deterministic ensemble networks using the Gaussian mean (purple curve), both modeling the transition dynamics in the source and target domains under determinism. To evaluate sample efficiency, we also include a policy trained

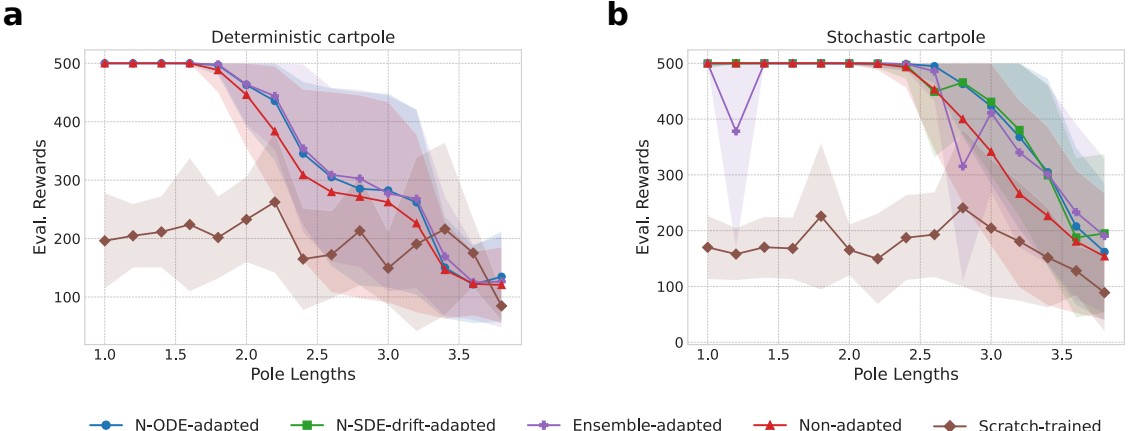

Figure 4: Evaluated performance of different policies in target cartpole environments with increasing pole lengths (source environment pole length = 1.0). Shaded regions denote the standard deviation of returns over 4 runs. **(a)** Under deterministic source and target dynamics, for pole lengths between 1.8 and 3.2, both the N-ODE–adapted and ensemble-adapted policies consistently outperform the original non-adapted policy, and both substantially surpass the trained-from-scratch policy. The N-ODE–adapted and ensemble-adapted policies show nearly identical performance. **(b)** Under stochastic source and target dynamics, obtained by adding zero-mean Gaussian noise to cart velocity, for pole lengths between 2.6 and 3.8, the ODE-adapted, SDE-drift–adapted, and ensemble-adapted policies achieve similarly strong performance, with the ensemble-adapted policy showing occasional drops at certain pole lengths. These are followed by the non-adapted policy, with the trained-from-scratch policy performing worst.

from scratch (brown curve) for 2k iterations, using the same number of target environment interactions as for training the augmented transition models. The results show that the two adapted policies perform similarly, consistently outperforming the non-adapted baseline, and remain significantly better than the scratch-trained policy when the gap between source and target environments is moderate. Although the scratch-trained policy is likely undertrained given the limited iterations, this comparison highlights that, with the same amount of target data, adapted policies perform substantially better, while a from-scratch policy would require far more samples to reach comparable performance. Performance of both adapted and non-adapted policies declines as pole length increases, likely because the limited action force cannot counteract the larger gravitational torque induced by a longer pole.

**Stochastic transition.** In the stochastic cartpole setting, both source and target transitions share the same level of stochasticity, with pole length as the only varying factor (Fig. 4b). We evaluate three adapted policies: ODE-adapted, SDE-drift–adapted, and ensemble-adapted (blue, green, and purple curves). These use, respectively, augmented neural ODEs, the augmented drift function of neural SDEs, and the mean of each augmented Gaussian network in the ensemble to model the averaged target dynamics. The ODE- and SDE-drift–adapted policies achieve nearly identical performance, indicating that in stochastic environments, the neural ODE effectively captures the same deterministic component as the SDE drift. The ensemble-adapted policy performs comparably, though with occasional drops likely due to suboptimal ensemble members. As in the deterministic case, all adapted policies outperform the non-adapted and scratch-trained baselines, though their performance also declines as pole length increases.

Interestingly, the source policy trained in a stochastic environment (red curve in Fig. 4b) is more robust to environmental changes than the one trained in a deterministic environment (red curve in Fig. 4a). This robustness likely arises because environmental stochasticity encourages broader exploration of the state–action space to counteract uncertainty, yielding a more generalizable policy. Consequently, adapted target policies benefit more from stronger source policies, as reflected by the larger performance gains over the non-adapted baseline in Fig. 4b compared to Fig. 4a.

## 4.3 Model-based policy learning

For more complex stochastic MuJoCo environments, we adopt a framework that interleaves exploration via MPC planning with model-free policy optimization, as described in Section 3.2. We compare the performance of policies derived from MPC planning using neural ODE/SDE models and their latent variants as transition models, under both full and partial observability. Additionally, we evaluate the family of neural differential equation–based policy optimization methods against a model-free baseline (SAC) and a model-based baseline (MBPO).

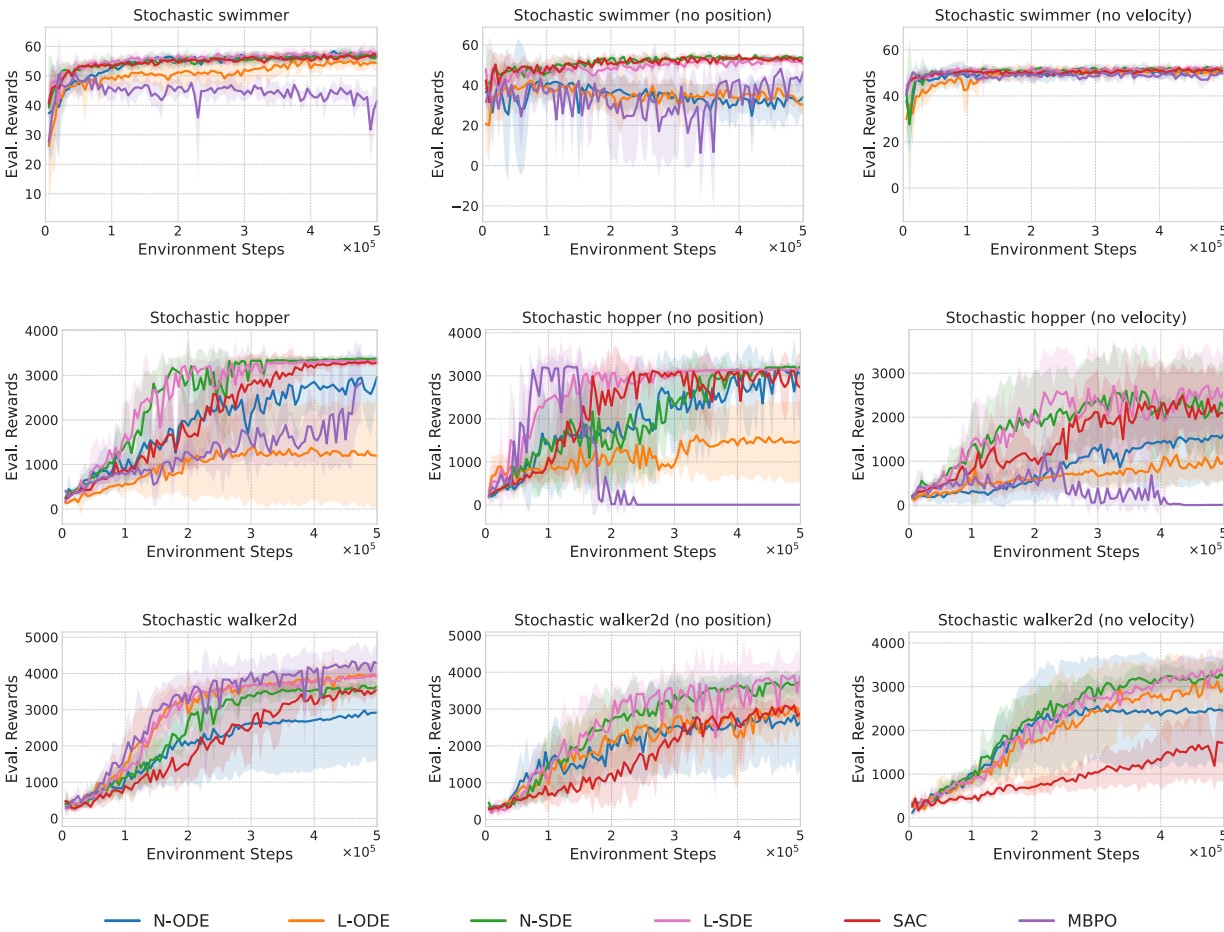

Figure 5: Learning curves of model-based and model-free policies in fully and partially observable stochastic Mujuco environments. Shaded regions indicate the standard deviation of evaluated returns over 5 runs, with evaluations conducted every 5k environment steps. The policy derived from the latent SDE model consistently achieves the best asymptotic performance across all environments. SDE-based policies also exhibit greater sample efficiency than the model-free baseline in more complex environments.

**SDE-based policies outperform their ODE-based counterparts.** Fig. 5 illustrates the learning curves for all baselines across different MuJoCo environments. SDE-based policies generally converge more rapidly and achieve superior solutions compared to ODE-based policies. The SDE-based models explicitly represent both the mean and variance of the stochastic transition dynamics, enabling more accurate prediction of the next (partial) observation than ODE-based models (Fig. 8 in Appendix B). This, in turn, allows SDE-based policies to better account for transition uncertainty and avoid high-risk actions. In contrast, ODE-based models capture only the mean dynamics. As a result, planning with them may deviate from the true environment trajectories, leading to suboptimal action choices and limiting exploration of new

state regions; policies trained on such transitions are therefore more prone to local minima during policy optimization.

**SDE-based policies achieve higher sample efficiency than model-free methods in sufficiently complex environments.**  In the stochastic hopper and walker2d environments (middle and bottom rows of Fig. 5), the policies with SDE-based planning demonstrate their advantage over the model-free baseline by requiring fewer interactions with the real environment (environment steps) to converge. This advantage is not observed in the simpler Swimmer environments. In particular, the model using latent SDE is the best performing in the POMDP of stochastic Hopper (no position). We further observe that the next partial observation predicted by the SDE transition in the latent space (latent SDE) aligns better with the real partial observations than those predicted by the SDE transition directly in the observation space (neural SDE) (see Fig. 9 in the Appendix B), which might explain the performance advantage.

**MBPO performs no better than SDE-based policies under stochastic transitions and fails under partial observability.**  While neural differential equation models have primarily shown advantages in modeling irregularly sampled time series (Rubanova et al., 2019; Kidger et al., 2020), here we also demonstrate their effectiveness in capturing stochastic dynamics within our model-based planning and policy learning framework, surpassing the MBPO baseline. In MBPO, transition noise can degrade model-generated samples, and the resulting errors may be exploited by the policy when trained on augmented buffers containing such samples. The slower convergence we observe in hopper is likely due to the capped, gradually increasing model horizon (see Appendix C): in stochastic settings, this leads to accumulated model errors, causing the model-generated rollouts to deviate from real trajectories and thereby degrading policy performance. By contrast, in the deterministic hopper, Janner et al. (2019) report that such a capped horizon accelerates learning without significantly deviating the trajectories. In stochastic walker2d, we use a single-step model horizon as in Janner et al. (2019), which avoids error accumulation from long horizons and yields performance comparable to the L-SDE baseline (left panel, bottom row of Fig. 5). However, MBPO completely fails in partially observable environments (e.g., stochastic Hopper without position or velocity; middle and right panels, middle row of Fig. 5). This failure arises in POMDP settings with early termination (such as hopper and walker2d), where the transition model cannot reliably predict termination signals from partial observations. As a result, model uncertainty is exploited by the policy, which converges to local minima corresponding to unhealthy state regions that trigger termination. In contrast, our model-based RL framework does not suffer from this issue in POMDPs, since models are used only for planning, while the policy is trained directly on environment transitions with real termination signals.

Finally, we note the computational overhead. SAC consistently requires the least training time, while ODE/SDE-based methods are slower—SDE models take longer than ODE models, and latent variants are slower than their state-based counterparts. MBPO is by far the most demanding. For instance, in stochastic Walker2d, N-ODE requires about $2\times$ the training time of SAC; L-ODE and N-SDE about $2\times$ that of N-ODE; L-SDE about $1.5\times$ that of N-ODE; and MBPO roughly $6\times$ longer than L-SDE. MBPO also consumes about $3\times$ more CPU memory than the other baselines. All experiments were run on a single NVIDIA A100 GPU. This overhead arises because ODE/SDE-based policies train an additional transition model compared to SAC, while MBPO is even more expensive due to the need to train and maintain an ensemble of models.

## 5  Discussion

We systematically evaluate action-conditional neural ODE and SDE models for policy adaptation and optimization in stochastic continuous-control environments. Policy adaptation via inverse dynamics, using neural ODE or SDE transition models, is substantially more sample-efficient than training from scratch when environment configurations change. While the original inverse dynamics work by Christiano et al. (2016) did not specify how the target transition is estimated or the training loss for the inverse model, our contribution is to make this explicit: we show how neural differential-equation transition models can be instantiated within the inverse-dynamics framework and contrasted with MBPO's ensemble transition models. Empirically, neural ODE/SDE models achieve adaptation performance comparable to the ensemble

model in a simple environment; more importantly, even in a stochastic domain where the change stems from a deterministic factor, a simple neural ODE matches a neural SDE with transformed drift.

For policy optimization, SDE-based models generally outperform other model-based and model-free approaches, highlighting the effectiveness of diffusion in modeling transition stochasticity. In particular, latent-SDE–based policies consistently exhibit strong sample efficiency and high asymptotic returns that are difficult to beat, suggesting the contribution of the latent space for robust policies under both full and partial observability. Moreover, using action planning to collect near-optimal transitions for policy training in our ODE/SDE-based framework provides clear advantages over the model-generated data used in MBPO, which tends to let the policy exploit model errors when transition dynamics or observations are noisy. These findings underscore the potential of policy planning over latent SDE for general robotic tasks subject to noisy transition dynamics and partial observability.

However, there are also situations where ODE/SDE–based policies may not offer advantages over standard model-based or model-free baselines. For instance, in continuous-control tasks with deterministic dynamics or very weak noise, ODE/SDE-based policies may not improve over model-based baselines such as MBPO. In stochastic walker2d, where the transition noise is relatively weak ($\mathcal{N}(0, 5^2)$ wind noise), MBPO with a single-step rollout horizon achieves performance comparable to the latent-SDE baseline. The slower convergence of MBPO in the stochastic Hopper arises instead from its capped and increasing rollout horizon: while this strategy improves sample efficiency in deterministic settings (Janner et al., 2019), it causes the policy to exploit model error in the stochastic case. By contrast, when the noise amplitude is large, as in the stochastic swimmer with $\mathcal{N}(0, 500^2)$ joint stiffness noise, MBPO even with a single-step horizon underperforms the SDE-based models.

A second limitation may arise in tasks without early termination conditions. Our results suggest that the sample-efficiency advantages of planning with ODE/SDE-based models are more pronounced relative to the model-free SAC in environments with termination conditions, where foresight helps agents avoid irreversible outcomes. Specifically, improved sample efficiency of SDE-based policies over SAC is observed only in tasks with termination conditions (stochastic hopper and walker2d), but not in tasks without them (stochastic swimmer). Nonetheless, alternative explanations for these differences are possible, and further investigation is required before drawing a conclusion.

Finally, there are task regimes where ODE/SDE-based models are inherently ill-suited. ODE/SDE continuous-time integrators struggle with hybrid or discontinuous dynamics, such as mode switches or resets, which degrade both planning and policy learning. In strongly chaotic regimes, small modeling errors can rapidly amplify under ODE/SDE rollouts, making one-step ensembles or model-free approaches more robust. Similarly, systems with actuation delays or dead zones are better described by delay differential equations or discrete-time history-based models (e.g., RNNs (Cho et al., 2014) or Transformers (Zerveas et al., 2021)) rather than by latent Markov ODE/SDE formulations.

### Acknowledgments

C. Han, S. Ioannou, A. Gilra, and E. Vasilaki acknowledge support from the CHIST-ERA project Causal Explanations in Reinforcement Learning (CausalXRL) (CHIST-ERA-19-XAI-002), funded by the Engineering and Physical Sciences Research Council (EPSRC), United Kingdom (grant EP/V055720/1). C. Han, S. Ioannou, L. Manneschi, M. Mangan, and E. Vasilaki acknowledge support from the EPSRC project Active Learning and Selective Attention for Robust, Transparent and Efficient AI (ActiveAI) (grant EP/S030964/1). C. Han, L. Manneschi, T. J. Hayward, and E. Vasilaki acknowledge support from the EPSRC project Magnetic Architectures for Reservoir Computing Hardware (MARCH) (grant EP/V006339/1).

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

# A   Stochastic cartpole environment

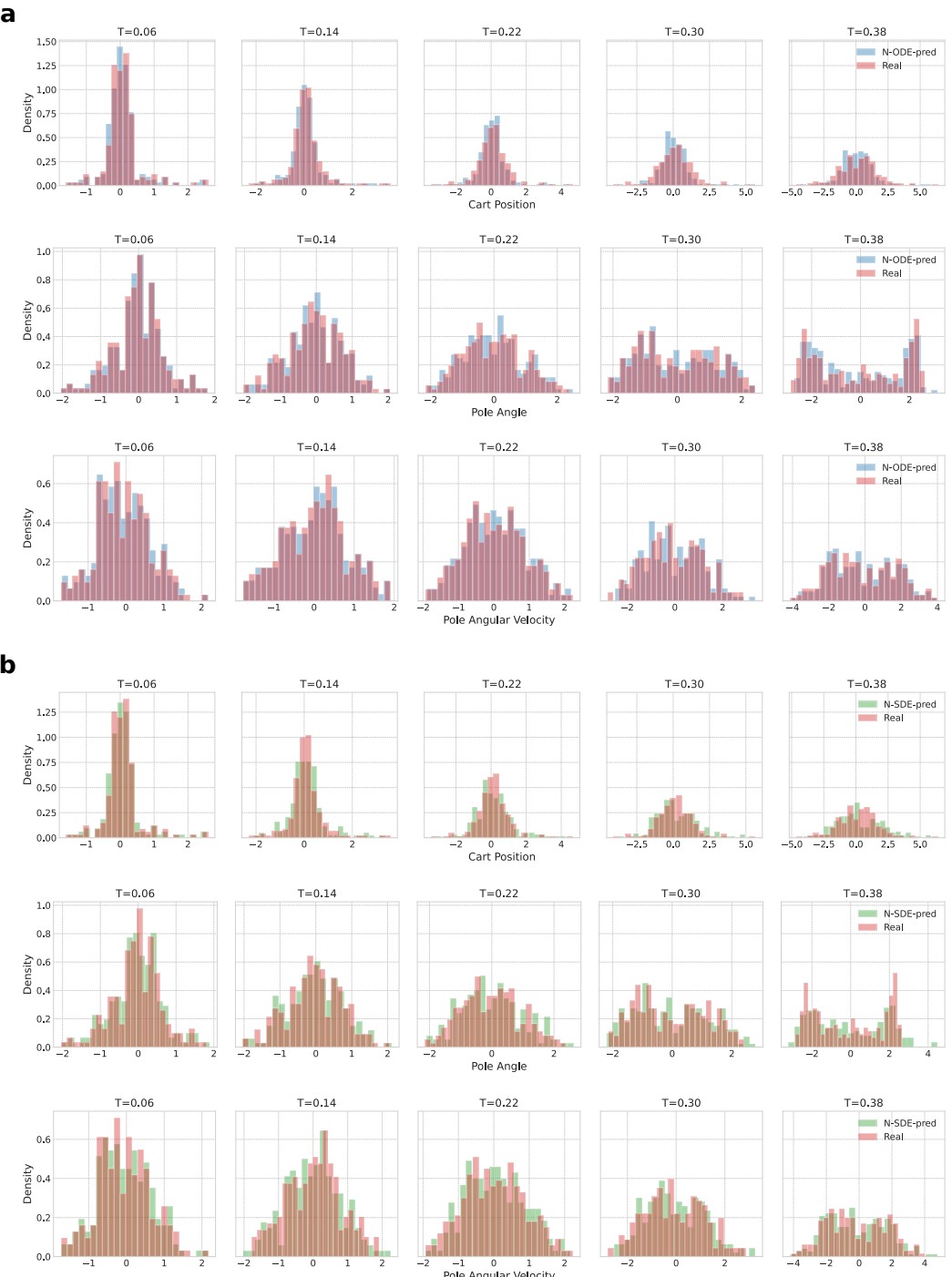

Figure 6: **(a)** Histograms of cart position, pole angle and pole angular velocity at 16%, 37%, 58%, 79%, 100% of the timesteps predicted by the neural ODE against the real histograms. **(b)** Similarly, histograms predicted by neural SDE against the real histograms.

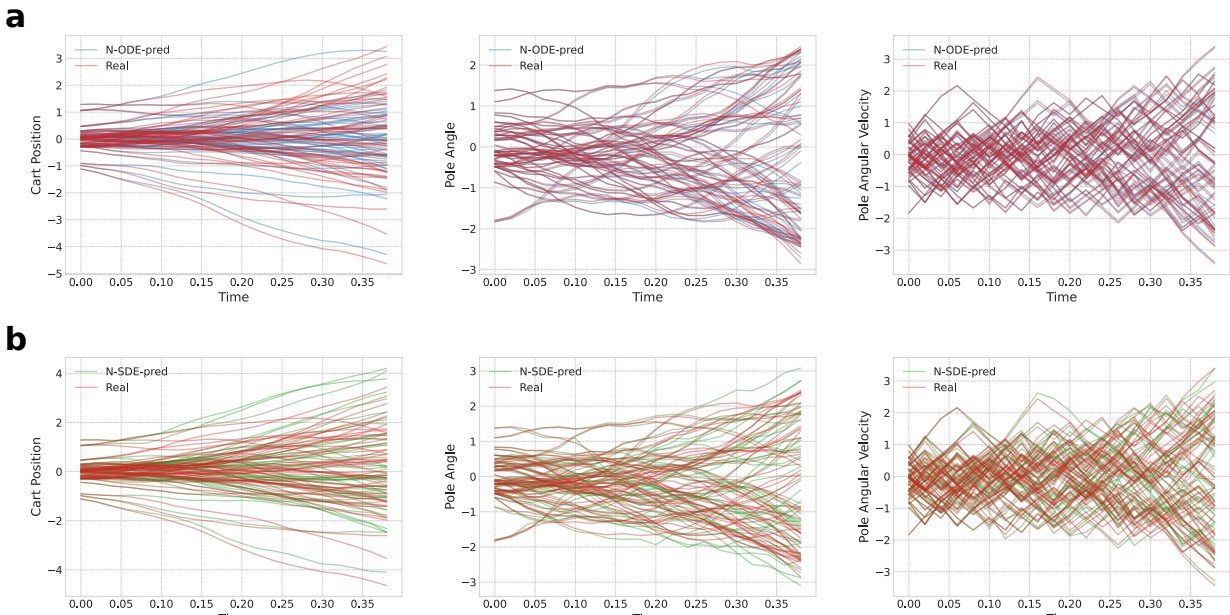

Figure 7: **(a)** 50 sample paths of cart position, pole angle and pole angular velocity predicted by neural ODE against 50 paths from the real distributions. **(b)** Similarly, paths predicted by neural SDE against paths from the real distributions. Since the transition dynamics of these features are deterministic, the neural ODE predicts almost the same distributions and their sample paths as the neural SDE does.

# B   Stochastic hopper environments

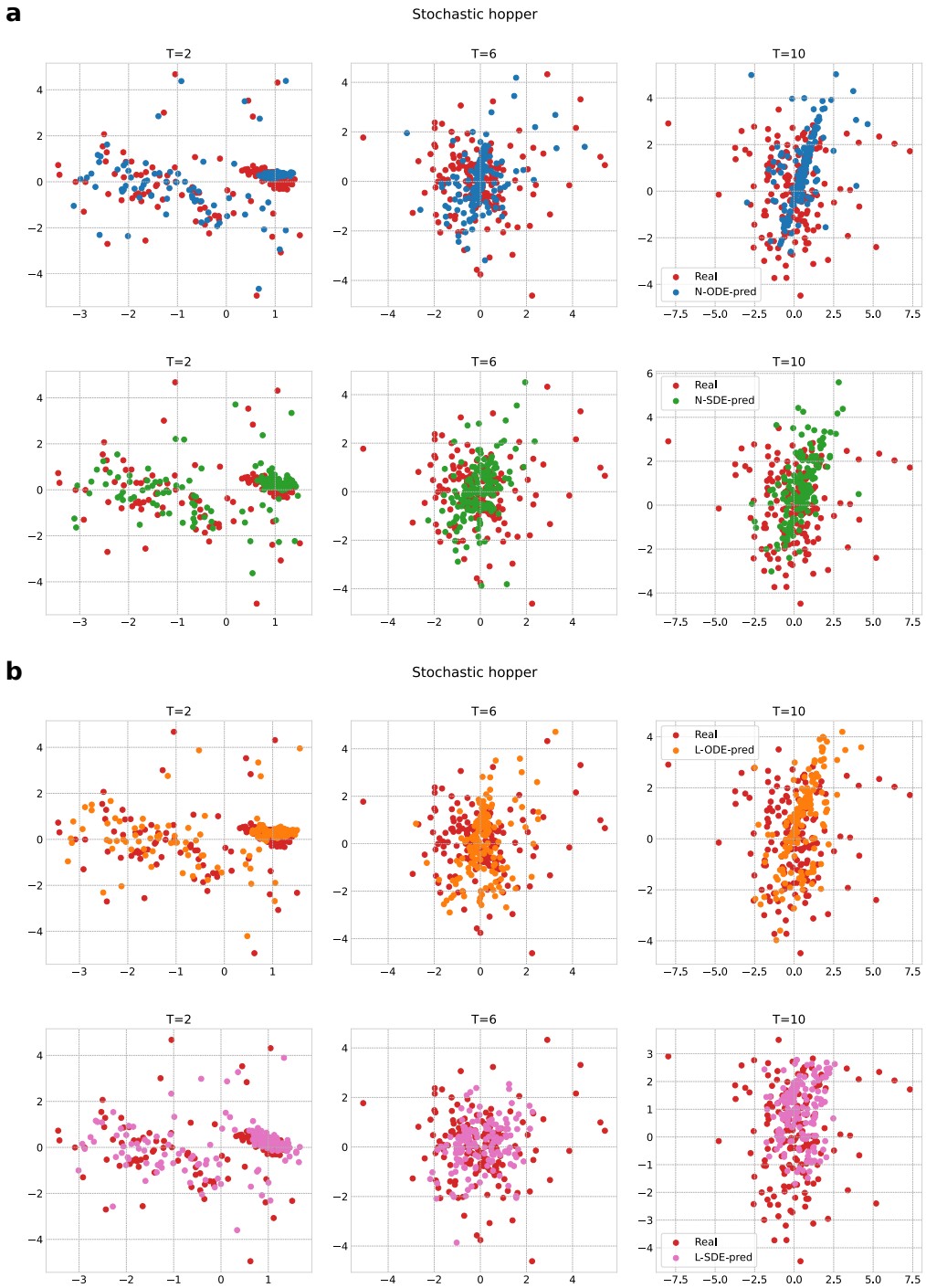

Figure 8: **(a)** Scatter plots of 200 PCA-embedded 2D representation of observations predicted by neural ODE and SDE, compared with that of the real observations across 20%, 60%, 100% of the timesteps in the stochastic hopper. **(b)** Similarly, scatter plots of 2D features predicted by latent ODE and SDE, against real ones. Predicted features by the SDE-based models demonstrate a better alignment with the real ones than those by the ODE-based models.

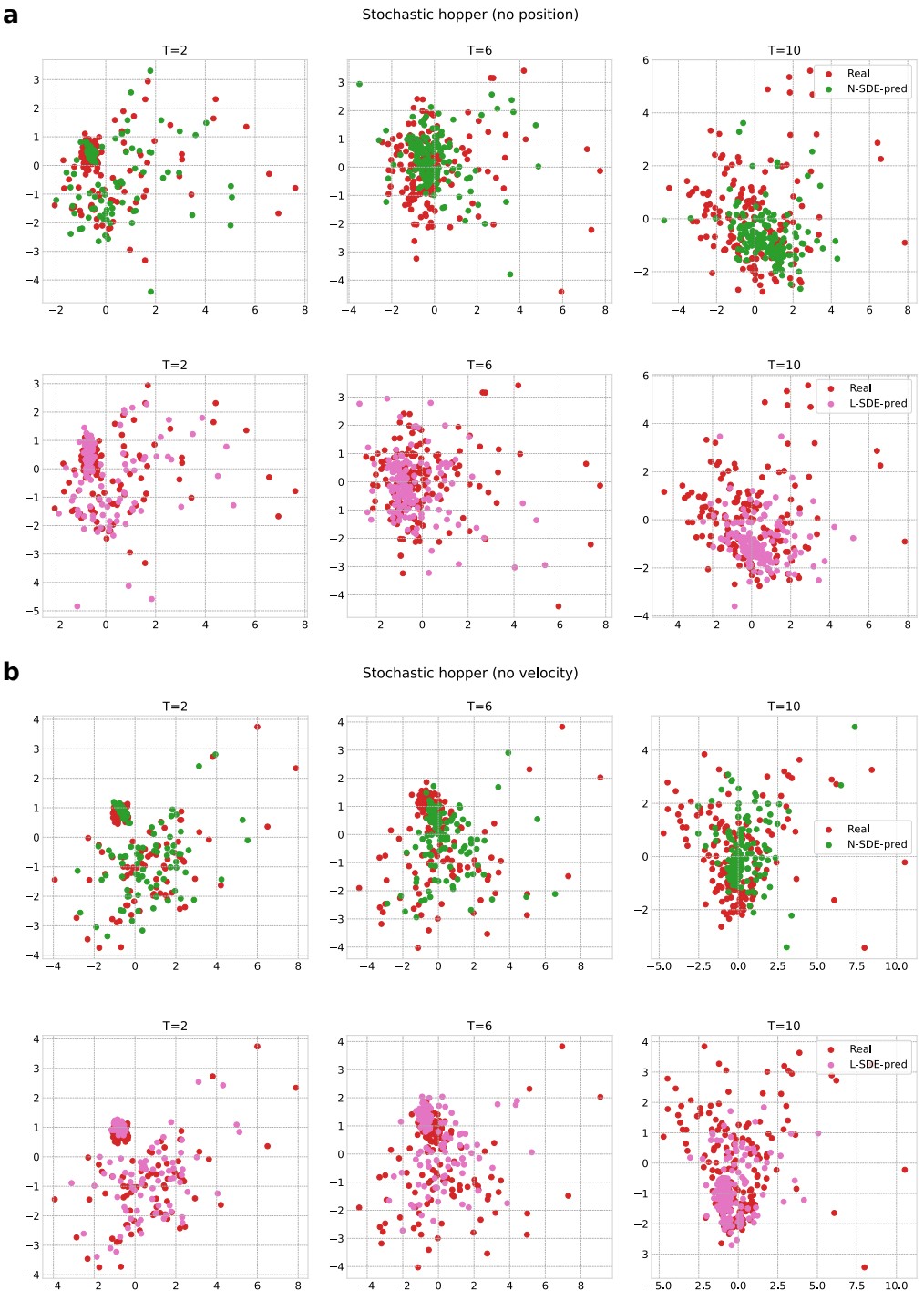

Figure 9: **(a)** Scatter plots of 200 PCA-embedded 2D representation of partial observations predicted by neural SDE and latent SDE, compared with that of the real partial observations, across 20%, 60%, 100% of the timesteps in the stochastic hopper without position observations. **(b)** Similar scatter plots in the stochastic hopper without velocity observations. Features predicted by the latent SDE align more closely with the real ones than those by the neural SDE.

## C  Experimental setup

Table 1: **Hyperparameter summary of model-based and model-free baselines across environments.**  Model architectures are described by network type and number of hidden layers with their sizes. For MBPO in Hopper, the model horizon follows a capped linear function of the epoch $i$: $f(i) = \min(\max(1 + 0.175 \times (i - 20), 1), 15)$, as in (Janner et al., 2019).

| | | Cartpole | Swimmer | Walker2d | Hopper |
|---|---|---|---|---|---|
| Training hparams | Max env. steps | 250K | | 500K | |
| | Model batch size | 256 | | 128 | |
| | Optimizer | | Adam | | |
| Neural/Latent ODE/SDE | Planning horizon | N/A | | 10 | |
| | Search population | N/A | 1000 | 1700 | |
| | Trajectory cut length | 20 | | 10 | |
| | ODE learning rate | 8e-4 | | 1e-3 | |
| | SDE generator learning rate | | 8e-4 | | |
| | SDE critic learning rate | 8e-5 | | 4e-5 | |
| | Encoder arch. | N/A | | GRU [128] | |
| | Decoder arch. | N/A | | Linear MLP | |
| | ODE (SDE drift func.) arch. | MLP [100, 100, 100, 100] | | MLP [100, 100, 100] | |
| | SDE diffusion func. arch. | MLP [32, 32] | | MLP [100, 100] | |
| | SDE critic arch. | MLP [100, 100, 100, 100, 100] | | MLP [100, 64, 64] | |
| | Solver | | Euler | | |
| SAC | Discount factor | | 0.99 | | |
| | Smoothing coef. | | 0.005 | | |
| | Learning rate | 1e-3 | | 3e-4 | |
| | Batch size | 32 | | 128 | |
| | Temperature | 0.3 | 0.2 | | 0.25 |
| | Actor/Critic arch. | | MLP [200, 200] | | |
| MBPO | Ensemble size | | 7 | | |
| | Model learning rate | | 1e-3 | | |
| | Model rollouts per env. step | | 400 | | |
| | Model horizon | | 1 | | Capped linear func. |
| | Model arch. | | MLP [200, 200, 200, 200] | | |

**Model training stopping criteria.**  For the Mujoco tasks, we apply early stopping to the ODE-based models to avoid overfitting: training is stopped if the MSE reconstruction error on the validation set does not decrease for $e$ consecutive epochs. Following Du et al. (2020), we use a linear decay schedule $e = \max(15 - i, 3)$, where $i$ is the epoch index in Algorithm 1. For the simpler carpole task, we train the ODE-based model for 50K iterations without early stopping, and terminate training of the Gaussian ensemble model when all members stop improving on the holdout MSE, as in Janner et al. (2019). The trained ODE-based model is then frozen and used as the drift component of the SDE-based models, while their additional components (the diffusion function of the SDE generator and the discriminator) are trained for 8K iterations.

**GAN-based training.**  We mitigate the instability of GAN-based SDE training by phasing the learning of the drift and diffusion components. First, we pre-train the drift as an ODE, optimized with MSE in the state space or ELBO in the latent space. We then freeze this drift and train the diffusion with a critic in a GAN-style setup. The ODE-based model is used as the drift since it empirically captures the mean dynamics of stochastic transitions, corresponding to the SDE drift function. Pretraining the drift substantially improves the stability of subsequent GAN-style training of the diffusion component. In addition, we found that ignoring actions in the critic input further stabilizes GAN training in the more complex Mujoco tasks.

