# OpenReview forum: "Neural ODE and SDE Models for Adaptation and Planning in Model-Based Reinforcement Learning"
_TMLR — Accepted by TMLR_

### Review · Reviewer_Pq2S · 2025-07-17

**Summary Of Contributions:**

The authors focus on model-based RL (MBRL) of a partially observable MDP with stochastic transitions.
They introduce an approach to model these dynamics in a latent-variable model by embedding
ordinary/stochastic differential equations (ODEs/SDEs) in its latent space.
Relying on a GAN-based approach. they are able to effectively adapt learned policies to new environments.
The authors evaluate their contributions in a series of detailed experiments.

**Audience:**

Yes

**Claims And Evidence:**

Yes

**Requested Changes:**

# Critical for acceptance recommendation
- A proper discussion on the experimental setup in the appendix.

# Nice to have
- A public implementation.

**Strengths And Weaknesses:**

## Strengths
- The method is novel and practically relevant.
- While the method consists of a relatively complex structure, the paper is reasonably well-written and well-explained.
- The empirical evaluation is thorough in evaluating each of the proposed contributions.

## Weaknesses
- The evaluation focuses primarily on demonstrating that the approach works and that the claims are valid. It mostly omits a comparison with prior work, i.e., lacks a comparison on whether it is a significant contribution to the literature.
- The model is rather complex.
- None of the experiments are reproducible as there is zero discussion on architecture, training process, optimizer, hyperparameters, etc..
- No public implementation is provided.

### Minor
- The GAN-based approach has the potential problem of being rather unstable, which is neither evaluated nor discussed.
- The assumption of deterministic observation emission seems relatively unrealistic for real-world applications and also not a requirement for the proposed model. While causing another source of randomness, the approach seems like it should be stable against it.
- There is no real ablation on the individual parts of the model.
- Red/green color coding in Figure 3 could pose a problem for some color-blind people. Should be double-checked with somebody who has it or with an online tool.

---

> ### Author Response · Authors · 2025-09-19
> **Responses to the reviewer**
>
> We thank the reviewer for the comments and have addressed them below. We are happy to respond to any further feedback.
>
> Comment 1:
> The evaluation focuses primarily on demonstrating that the approach works and that the claims are valid. It mostly omits a comparison with prior work, i.e., lacks a comparison on whether it is a significant contribution to the literature.
>
> Response 1:
> We have strengthened comparisons with prior work by adding ensemble-based adaptation baselines and MBPO for Mujoco tasks, including the more challenging Walker2d benchmark. Our results show that latent-SDE–based policies consistently demonstrate strong sample efficiency and high asymptotic returns, highlighting their applicability to continuous-control tasks with noisy dynamics and partial observability. We also refer the reviewer to our Response 1 addressing a similar concern raised in Comment 1 by reviewer gsrz.
>
> Comment 2:
> None of the experiments are reproducible as there is zero discussion on architecture, training process, optimizer, hyperparameters, etc..
> Critical for acceptance recommendation: a proper discussion on the experimental setup in the appendix.
>
> Response 2:
> We have now added a detailed description of the experimental setup in Appendix C.
>
> Comment 3:
> No public implementation is provided. Nice to have: a public implementation.
>
> Response 3:
> Upon acceptance, we will release a GitHub repository with the full code. We prefer not to release at this stage to avoid compromising author anonymity.
>
> Minor:
>
> Comment 4:
> The GAN-based approach has the potential problem of being rather unstable, which is neither evaluated nor discussed.
>
> Response 4:
> We mitigate the instability of GAN-based SDE training by phasing the learning of the drift and diffusion components. First, we pre-train the drift as an ODE, optimized with MSE in the state space or ELBO in the latent space. We then freeze this drift and train the diffusion with a discriminator in a GAN-style setup. The ODE-based model is used as the drift since it empirically captures the mean dynamics of stochastic transitions, corresponding to the SDE drift function. Pretraining the drift substantially improves the stability of subsequent GAN-style training of the diffusion component.  In addition, we found that ignoring actions in the critic input further stabilizes GAN training in the more complex Mujoco tasks. We have included the above discussion in the last paragraph of Appendix C.
>
> Comment 5:
> The assumption of deterministic observation emission seems relatively unrealistic for real-world applications and also not a requirement for the proposed model. While causing another source of randomness, the approach seems like it should be stable against it.
>
> Response 5:
> To specifically evaluate the model’s ability to capture stochasticity in the transition dynamics, and to decouple it from stochasticity in the observation emission, we simplified the setting to consider only transition noise as the source of aleatoric uncertainty. However, our approach readily generalizes to noisy observation emissions by replacing the deterministic observation model with a probabilistic neural network that outputs the statistics of the predicted observation distribution. This clarification has been added to the second paragraph of Section 2.
>
>
> Comment 6:
> There is no real ablation on the individual parts of the model.
>
> Response 6:
> Although not explicitly labeled as ablations, our experiments include comparisons that isolate the contributions of individual model components. Specifically, a neural SDE reduces to a neural ODE when its additive diffusion component is removed. A latent ODE reduces to a neural ODE when the RNN encoder and observation decoder (which map states into and out of latent space) are removed. Similarly, a latent SDE reduces to a neural ODE when both the encoder–decoder pair and the diffusion component are removed. By comparing neural ODE/SDE models with their latent variants in stochastic Mujoco tasks, we demonstrate the effectiveness of each additional component: the diffusion function for capturing transition stochasticity, and the latent space for enabling robust policy learning under both full and partial observability.

---

> > ### Comment · Reviewer_Pq2S · 2025-09-23
> >
> > Thank you for the response.
> >
> > It could be that I missed it in one of the answers, but can the authors explain what happened in the environmental setups?
> >
> > - The noise for the swimmer exploded from an already large $N(0,10^4)$ to $N(0,500^2)$.
> > - The noise for the stochastic hopper, on the other hand, collapsed from $N(0,10^6)$ to $N(0,5^2)$.
> >
> > But in both scenarios, the reported results remained identical, as far as I can tell?

---

> > > ### Author Response · Authors · 2025-09-23
> > > **Responses to the reviewer**
> > >
> > > We thank the reviewer for pointing this out. The noise values for Swimmer and Hopper were mistakenly reported in the first version of the draft. We have corrected them in the current version to match those used in our code: $\mathcal{N}(0, 500^2)$ for Swimmer and $\mathcal{N}(0, 5^2)$ for Hopper. The results reported in the initial draft were already based on these values, which is why the results remained the same across versions.

---

### Review · Reviewer_JiwX · 2025-07-20

**Summary Of Contributions:**

This paper proposes neural ODE and SDE based dynamics models for model based RL and adaptation in changing environments. Environments with both full and partial observability with removed/masked out feature dimension are considered. Adaptation is based on predicting desired next state in the original environment and training an inverse dynamics model to find an action in the target environment. Experiments are conducted in the stochastic and masked versions of cartpole, swimmer, and hopper. It was shown that in stochastic environments, neural SDE better captures environment stochasticity than neural ODE. The inverse dynamics model based adaptation strategy significantly outperforms training from scratch. The authors also conducted regular model based RL experiments, using a model free algorithm as baseline and ODE vs SDE and latent vs no-latent as ablation. It was shown that latent SDE performs best overall.

**Audience:**

Yes

**Claims And Evidence:**

No

**Requested Changes:**

* The author should clarify that they consider discrete time MDP, if that's the case, and describe why experimenting neural ODE and SDE in this setting is important.
* The authors should add a regular model-based RL baseline as a comparison for both planning and adaptation, or argue why such comparison is not needed.
* "Empirically we only require a small amount of data from the target domain to train the target transition
model augmented from the source transition model, compared with training a target transition model from
scratch." Do you have examples or empirical data to show this. For example, how many transitions do you need to train the target transition model from scratch?
* "We freeze the target transition model once it is sufficiently accurate" How accurate is sufficiently accurate? What's your model training stopping criteria?
* In Figure 4, when you train a policy from scratch for each pole length, how many training steps did you use? It's hard for readers to know if the from scratch policy was sufficiently trained. Generally cartpole is an easy environment and we would expect most RL algorithms to work no matter the pole length.
* "In contrast, ODE-based models capture only the mean dynamics, and policies planned on them may be more prone to local minima" Could you expand on why ODE based models would be more prone to local minima? Is it the model optimization local minima or the policy optimization local minima?

**Strengths And Weaknesses:**

**Strength**
* This paper does a good job describing neural ODE and SDE, which may not be familiar to all RL readers. The authors also included important references such as Chen et al, 2018, Du et al, 2020 for readers to get familiar with the subject.
* The experiments and adaptation strategy are described in (almost) sufficient detail to be reproduced.

**Weakness**
* Although not explicitly called out, I believe this paper consider discrete time MDP rather than continuous time MDP as in Du et al, 2020. Why is using neural ODE and SDE important in this setting, as neural ODE is usually argued to be only stronger in irregularly sampled time series?
* In the same spirit as above, I believe a model-based RL baseline is missing from all experiments. A model-based RL baseline such as MBPO, PETS, or dreamer (for POMDP) can also be applied in the same inverse dynamics model based adaptation strategy.
* Some parts of the methods and experiments section can be more detailed. I describe these below.

---

> ### Author Response · Authors · 2025-09-19
> **Responses to the reviewer - Part 1**
>
> We thank the reviewer for the comments and have addressed them below. We are happy to respond to any further feedback.
>
> Comment 1:
> Although not explicitly called out, I believe this paper consider discrete time MDP rather than continuous time MDP as in Du et al, 2020. Why is using neural ODE and SDE important in this setting, as neural ODE is usually argued to be only stronger in irregularly sampled time series?
> The author should clarify that they consider discrete time MDP, if that's the case, and describe why experimenting neural ODE and SDE in this setting is important.
>
> Response 1:
> We indeed consider a discrete-time MDP framework, in contrast to the continuous-time SMDP setting studied in Du et al. (2020). In our case, neural ODE/SDE models are used as continuous-time dynamics to fit discrete-time data points. This is important because, unlike general model-based RL baselines such as MBPO and PETS, which operate strictly at a fixed environment simulator step, neural ODE/SDE models in principle allow integrator step sizes smaller than the environment step. This enables finer temporal resolution and better error control (albeit with higher computational cost). For efficiency, we set the integrator step size equal to the simulator step size, but we emphasize this added flexibility of neural ODE/SDE models. We have clarified this point in the paragraph preceding Section 4.1.
> In addition, while neural differential equations are often motivated by irregularly sampled time series, we also show their effectiveness in capturing stochastic dynamics within our model-based planning and policy learning framework, where they outperform the MBPO baseline (see Section 4.3).
>
> Comment 2:
> In the same spirit as above, I believe a model-based RL baseline is missing from all experiments. A model-based RL baseline such as MBPO, PETS, or dreamer (for POMDP) can also be applied in the same inverse dynamics model based adaptation strategy.
> The authors should add a regular model-based RL baseline as a comparison for both planning and adaptation, or argue why such comparison is not needed.
>
> Response 2:
> We have strengthened comparisons with prior work by adding ensemble-based adaptation baselines and MBPO for Mujoco tasks, including the more challenging Walker2d benchmark. The results show that latent-SDE–based policies consistently achieve strong sample efficiency and high asymptotic returns, underscoring their applicability to continuous-control tasks with noisy dynamics and partial observability. We also refer the reviewer to our Response 1 to Comment 1 by reviewer gsrz, which raised a similar concern.
>
> Comment 3:
> "Empirically we only require a small amount of data from the target domain to train the target transition model augmented from the source transition model, compared with training a target transition model from scratch." Do you have examples or empirical data to show this. For example, how many transitions do you need to train the target transition model from scratch?
>
> Response 3:
> In the cartpole adaptation task with varying pole lengths, training a neural ODE transition model from scratch requires 500K environment transitions collected with a random policy. By contrast, when transferring from the source to the target environment, we only fine-tune the final layer of the neural ODE while keeping the earlier layers fixed from the source-trained model. This fine-tuning requires just 2K target transitions to reach the same order of final validation loss as training from scratch. This clarification has been added to the paragraph immediately following Eq. 16.
>
> Comment 4:
> "We freeze the target transition model once it is sufficiently accurate" How accurate is sufficiently accurate? What's your model training stopping criteria?
>
> Response 4:
> For the Mujoco tasks, we apply early stopping to the ODE-based models to avoid overfitting: training is stopped if the MSE reconstruction error on the validation set does not decrease for $e$ consecutive epochs. Following Du et al. (2020), we use a linear decay schedule $e=\text{max}(15-i, 3)$, where $i$ is the epoch index in Algorithm 1. For the simpler carpole task, we train the ODE-based model for 50K iterations without early stopping, and terminate training of the Gaussian ensemble model when all members stop improving on the holdout MSE, as in Janner et al. (2019). The trained ODE-based model is then frozen and used as the drift component of the SDE-based models, while their additional components (the diffusion function of the SDE generator and the discriminator) are trained for 8K iterations. The training details are provided in Appendix C.

---

> > ### Comment · Reviewer_JiwX · 2025-09-20
> >
> > Thank the authors for their responses. A lot of my questions are resolved.
> >
> > A few follow-ups:
> > * I did not quite understand the purpose of the adaptation experiments, in relation to reviewer gsrz's point on demonstrating when neural ODE/SDE works and when it doesn't. Other than showing that adaptation and fine-tuning works better than training from scratch, we can't really make other conclusions from the experiments other than that the ODE/SDE dynamics models perform similarly to MLP dynamics models in MBPO?
> > * Can you say a bit more about your MBPO implementation? How much is it consistent with Janner's implementation or other implementations online? In the MBPO paper, they reached near optimal performance with about 50k steps in Hopper, whereas your MBPO barely reached the same performance in 500k steps. I understand there is some difference between Janner's experiments in deterministic Hopper, whereas your Hopper is stochastic. But i think the difference is small?

---

> > > ### Author Response · Authors · 2025-09-21
> > > **Responses to the reviewer**
> > >
> > > We thank the reviewer for the comments and have addressed the follow-up questions below. We are happy to provide further clarification if needed.
> > >
> > > Comment 1:
> > > I did not quite understand the purpose of the adaptation experiments, in relation to reviewer gsrz's point on demonstrating when neural ODE/SDE works and when it doesn't. Other than showing that adaptation and fine-tuning works better than training from scratch, we can't really make other conclusions from the experiments other than that the ODE/SDE dynamics models perform similarly to MLP dynamics models in MBPO?
> > >
> > > Response 1:
> > > In the original inverse dynamics paper by Christiano et al. (2016), the authors did not specify how the target transition is estimated or the detailed training loss of the inverse model. Our aim and contribution here are to make this explicit: we demonstrate how neural differential equation–based transition models can be applied within the inverse dynamics framework for policy adaptation, in comparison with MBPO’s ensemble transition models. The results show that neural ODE/SDE models achieve similar adaptation performance to the ensemble model in a simple environment. More importantly, even in a stochastic domain where the change arises from a deterministic factor, a simple neural ODE can perform as well as a neural SDE with transformed drift. We have added this clarification in  Section 5 Conclusion.
> > >
> > > Comment 2:
> > > Can you say a bit more about your MBPO implementation? How much is it consistent with Janner's implementation or other implementations online? In the MBPO paper, they reached near optimal performance with about 50k steps in Hopper, whereas your MBPO barely reached the same performance in 500k steps. I understand there is some difference between Janner's experiments in deterministic Hopper, whereas your Hopper is stochastic. But i think the difference is small?
> > >
> > > Response 2:
> > > Our MBPO implementation is a PyTorch replica of Janner’s original TensorFlow code, following the public repository https://github.com/Xingyu-Lin/mbpo_pytorch. We adopt the same hyperparameter values as in Appendix C of Janner et al., except that we use a smaller model batch size of 128 (instead of 256) to match the neural differential equation models. The slower convergence we observe in Hopper is likely due to the capped increasing model horizon: in stochastic settings this introduces accumulated model errors, causing the model-generated rollouts to deviate from real trajectories and degrading policy performance. By contrast, in deterministic Hopper, Janner et al. report that such a capped horizon accelerates learning without significantly deviating the trajectories. In Walker2d, we use a model horizon of 1 as in Janner et al., which avoids error accumulation from long horizons and yields performance comparable to the latent-SDE model. This discussion has also been added to the last paragraph of Section 4.3.

---

> > > > ### Comment · Reviewer_JiwX · 2025-09-24
> > > >
> > > > Thank the authors for further explanation. I have no more questions.

---

> ### Author Response · Authors · 2025-09-19
> **Responses to the reviewer - Part 2**
>
> Comment 5:
> In Figure 4, when you train a policy from scratch for each pole length, how many training steps did you use? It's hard for readers to know if the from scratch policy was sufficiently trained. Generally cartpole is an easy environment and we would expect most RL algorithms to work no matter the pole length.
>
> Response 5:
> We trained the target model-free policy from scratch for 2k iterations, using the same number of target environment interactions as for training the augmented target transition models. We acknowledge that this from-scratch policy is likely undertrained given the limited iterations. Our intention, however, is to highlight that with the same amount of target data, adapted policies perform significantly better, while a from-scratch policy would require many more samples to reach comparable performance. We have added this explanation in the second paragraph of Section 4.2.
>
> Comment 6:
> "In contrast, ODE-based models capture only the mean dynamics, and policies planned on them may be more prone to local minima" Could you expand on why ODE based models would be more prone to local minima? Is it the model optimization local minima or the policy optimization local minima?
>
> Response 6:
> We are referring to local minima in policy optimization. Since ODE-based models cannot capture the stochasticity of the transition dynamics, planning with them may deviate from the true environment trajectories. As a result, the selected actions can be suboptimal for exploring new state regions, and policies trained on these suboptimal transitions are more likely to get trapped in local minima. The text has been updated to incorporate this explanation.

---

### Review · Reviewer_gsrz · 2025-08-03

**Summary Of Contributions:**

The authors present a study of neural ODE and SDE applications to model-based RL. They argue these are powerful tools to model dynamical systems since they restrict the model complexity to a physics informed class that should be able to learn better models for RL. They first demonstrate how these models result in better adaptation to changes in the environment, and then propose a framework to use neural ODE/SDE in latent space, and test these methods in a set of RL tasks.

**Audience:**

Yes

**Broader Impact Concerns:**

No concerns.

**Claims And Evidence:**

Yes

**Requested Changes:**

I would like to see a deeper empirical evaluation. I would be very interested in seeing how well the methods proposed compare to other state of the art model-based RL baselines. Please note that I do not expect the proposed method to beat any benchmark at any task. I simply believe that a thorough study of when does using neural ODE/SDE methods make sense, versus when it is perhaps not necessary or not valuable, would greatly increase the quality and impact of the work. I understand the authors make an attempt to this with the environments considered, but in my opinion these are not really sufficient.

I would expect neural ODE/SDE methods to be more sample efficient, but also more computationally expensive in some instances. There are problems where perhaps modelling the dynamics as a conditioned gaussian in latent space (like eg MBPO) would suffice, but I assume in other environments it would not. I also believe that the complexity constrained nature of physics based models should yield improvements across other robotics tasks, which are sometimes notoriously difficult for model-based RL.

In summary, I would like the work to provide a more thorough set of results providing intuition on the utility and necessity for ODE/SDE models in model-based RL.

**Strengths And Weaknesses:**

## Strengths
- I believe that physics constrained models are very relevant for model-based RL, and the work does a good job at introducing the use of neural SDEs in latent space, which seems like a powerful approach.
- The proposed framework splitting the latent dynamics learning and the stochasticity modelling via a GAN seems novel and is intuitive.
- The paper is clear, well written and provides a good cover of other ODE/SDE learning methods.

## Weaknesses
- The main weakness is the weak empirical evaluation. Given that the paper does not derive any novel theoretical results, I would expect a thorough study on more diverse environments, ablations between models, comparisons between ODE and SDE approaches and, perhaps more importantly, comparisons between ODE/SDE methods and other general model based and model free RL approaches. I think the paper could be a very good contribution if it would attempt to answer, more thoroughly, the question of "when are ODE/SDE models necessary, and when are they not?".

---

> ### Author Response · Authors · 2025-09-19
> **Responses to the reviewer**
>
> We thank the reviewer for the comments and have addressed them below. We are happy to respond to any further feedback.
>
> Comment 1:
> The main weakness is the weak empirical evaluation. Given that the paper does not derive any novel theoretical results, I would expect a thorough study on more diverse environments, ablations between models, comparisons between ODE and SDE approaches and, perhaps more importantly, comparisons between ODE/SDE methods and other general model based and model free RL approaches. I think the paper could be a very good contribution if it would attempt to answer, more thoroughly, the question of "when are ODE/SDE models necessary, and when are they not?".
>
> I would like to see a deeper empirical evaluation. I would be very interested in seeing how well the methods proposed compare to other state of the art model-based RL baselines. Please note that I do not expect the proposed method to beat any benchmark at any task. I simply believe that a thorough study of when does using neural ODE/SDE methods make sense, versus when it is perhaps not necessary or not valuable, would greatly increase the quality and impact of the work. I understand the authors make an attempt to this with the environments considered, but in my opinion these are not really sufficient.
>
> I would expect neural ODE/SDE methods to be more sample efficient, but also more computationally expensive in some instances. There are problems where perhaps modelling the dynamics as a conditioned gaussian in latent space (like eg MBPO) would suffice, but I assume in other environments it would not. I also believe that the complexity constrained nature of physics based models should yield improvements across other robotics tasks, which are sometimes notoriously difficult for model-based RL.
>
> In summary, I would like the work to provide a more thorough set of results providing intuition on the utility and necessity for ODE/SDE models in model-based RL.
>
> Response 1:
> We have strengthened the empirical evaluation by incorporating the following new simulations:
>
> 1. Adaptation task. We introduced an adapted policy baseline using the ensemble of Gaussian neural networks from MBPO/PETS. In deterministic settings, ODE- and ensemble-adapted policies perform similarly and clearly outperform both non-adapted and scratch-trained baselines. In stochastic settings, ODE- and SDE-drift–adapted policies show nearly identical performance, while the ensemble baseline performs comparably with occasional drops. Overall, all adapted policies outperform non-adapted and scratch-trained baselines. Please see the updated Section 4.2 for details.
>
> 2. Policy learning in MuJoCo tasks. We added MBPO as a baseline and included the more challenging Walker2d benchmark. Results show that SDE-based models capture transition stochasticity more effectively, with latent-SDE policies consistently achieving strong sample efficiency and high asymptotic returns under both full and partial observability. These findings illustrate the applicability of latent-SDE–based planning for robotic tasks subject to noisy transition dynamics and partial observability. Please see the updated Section 4.3 and Section 5 Conclusion.
>
> Overall, our results highlight the effectiveness of the latent SDE model as a transition model in continuous-control tasks with stochastic dynamics, where the diffusion function captures transition stochasticity and the latent space enables robust policy learning under both full and partial observability. Moreover, using action planning to collect near-optimal transitions for policy training in our ODE/SDE-based framework provides clear advantages over the model-generated data used in MBPO, which tends to let the policy exploit model errors when transition dynamics or observations are noisy.

---

> > ### Comment · Reviewer_gsrz · 2025-09-23
> >
> > I thank the authors for addressing my main concern. The new experiments and discussion help understand better the impact of the SDE models.
> >
> > I have one last question. Is there some problem instance where the authors estimate that latent neural ODE/SDE methods would not provide any improvement over more standard model based / model free RL, or would even perform worse?
> >
> > (and related, what was the computational overhead of the proposed method with respect to these? Authors mention MBPO yielding higher compute time, but what about latent SDE vs standard RL algorithms, like SAC, which seems to produce similar performance in some tasks?)

---

> > > ### Author Response · Authors · 2025-09-24
> > > **Responses to the reviewer**
> > >
> > > We thank the reviewer for the thoughtful follow-up. There are indeed situations where (latent) neural ODE/SDE–based policies may not offer advantages over standard model-based or model-free baselines. For instance, in continuous-control tasks with deterministic dynamics or very weak noise,  ODE/SDE-based policies may not improve over model-based baselines such as MBPO. In stochastic walker2d, where the transition noise is relatively weak ($\mathcal{N}(0, 5^2)$ wind noise), MBPO with a single-step rollout horizon achieves performance comparable to the latent-SDE baseline. The slower convergence of MBPO in the stochastic Hopper arises instead from its capped and increasing rollout horizon: while this strategy improves sample efficiency in deterministic settings (Janner et al., 2019), it causes the policy to exploit model error in the stochastic case. By contrast, when the noise amplitude is large, as in the stochastic swimmer with $\mathcal{N}(0, 500^2)$ joint stiffness noise, MBPO even with a single-step horizon underperforms the SDE-based models.
> > >
> > > A second limitation may arise in tasks without early termination conditions. Our results suggest that the sample-efficiency advantages of planning with ODE/SDE-based models are more pronounced relative to the model-free SAC in environments with termination conditions, where foresight helps agents avoid irreversible outcomes. Specifically, improved sample efficiency of SDE-based policies over SAC is observed only in tasks with termination conditions (stochastic hopper and walker2d), but not in tasks without them (stochastic swimmer). Nonetheless, alternative explanations for these differences are possible, and further investigation is required before drawing a conclusion.
> > >
> > > Beyond these examples, there are also specific settings where ODE/SDE-based models are ill-suited. ODE/SDE continuous-time integrators struggle with hybrid or discontinuous dynamics, such as mode switches or resets, which degrade both planning and policy learning. In strongly chaotic regimes, small modeling errors can rapidly amplify under ODE/SDE rollouts, making one-step ensembles or model-free approaches more robust. Similarly, systems with actuation delays or dead zones are better described by delay differential equations or discrete-time history-based models (e.g., RNNs or Transformers) rather than by latent Markov ODE/SDE.
> > >
> > > With respect to computational overhead, we find that SAC consistently requires the least training time. ODE/SDE-based methods are slower, with SDE models taking longer than ODE models and latent variants slower than their state-based counterparts. MBPO is the most computationally demanding. For instance, in stochastic walker2d, N-ODE requires about 2× the training time of SAC; L-ODE and N-SDE take about 2× as long as N-ODE; L-SDE takes about 1.5× longer than N-ODE; and MBPO requires roughly 6× longer than L-SDE. Moreover, MBPO consumes about 3× more CPU memory than the other baselines. This overhead arises because ODE/SDE-based policies train an additional transition model compared to SAC, while MBPO is even more expensive due to the need to train and maintain an ensemble of models.
> > >
> > > These discussions have been incorporated into Section 5.

---

> > > > ### Comment · Reviewer_gsrz · 2025-09-24
> > > >
> > > > I thank the authors for this detailed reply. I don't have any further questions.

---

### Decision · Action_Editor_MaV6 · 2025-09-28

**Recommendation:** Accept as is

**Additional Comments:**

The paper has solid contributions, and is well written.

**Audience:**

Yes

**Audience Explanation:**

The paper proposes a neural neural ODE and SDE framework for model-based RL (MBRL). The work is of interest to communities of RL and robotics, among others.

**Claims And Evidence:**

Yes

**Claims Explanation:**

The paper studies neural ODE and SDE approaches to model-based RL in fully and partially observable environments. The proposed framework is empirically validated in benchmarks, yielding improvements over strong MBRL baselines. There were concerns with evaluation and baselines in the original submission. The revised version made substantial improvements, and all reviewers were satisfied by the empirical evidence.